# In Search of Smooth Minima for Purifying Backdoor in Deep Neural Networks

## Abstract

The success of a deep neural network (DNN) heavily relies on the details of the training scheme; *e.g.*, training data, architectures, hyper-parameters, *etc.* Recent backdoor attacks suggest that an adversary can take advantage of such training details and compromise the integrity of a DNN. Our studies show that a backdoor model is usually optimized to a *bad local minima*, *i.e.*, sharper minima as compared to a benign model. Intuitively, a backdoor model can be purified by re-optimizing the model to a smoother minima through fine-tuning with a few clean validation data. However, fine-tuning all DNN parameters often requires huge computational cost and often results in sub-par clean test performance. To address this concern, we propose a novel backdoor purification technique—Natural Gradient Fine-tuning (NGF)—which focuses on removing backdoor by fine-tuning *only one layer*. Specifically, NGF utilizes a loss surface geometry-aware optimizer that can successfully overcome the challenge of reaching a smooth minima under a one-layer optimization scenario. To enhance the generalization performance of our proposed method, we introduce a clean data distribution-aware regularizer based on the knowledge of loss surface curvature matrix, *i.e.*, *Fisher Information Matrix*. Extensive experiments show that the proposed method achieves state-of-the-art performance on a wide range of backdoor defense benchmarks: *four different datasets—CIFAR10, GTSRB, Tiny-ImageNet, and ImageNet; 13 recent backdoor attacks, e.g., Blend, Dynamic, WaNet, ISSBA, etc.*

## 1 Introduction

Training a deep neural network (DNN) with a fraction of poisoned or malicious data is often security-critical since the model can successfully learn both clean and adversarial tasks equally well. This is prominent in scenarios where one outsources the DNN training to a vendor. In such scenarios, an adversary can mount backdoor attacks (Gu et al., 2019; Chen et al., 2017) through poisoning a portion of training samples so that the model will misclassify any sample with a *particular trigger* or *pattern* to an adversary-set label. Whenever a DNN is trained in such a manner, it becomes crucial to remove the effect of backdoor before deploying it for a real-world application.

Different defense techniques (Liu et al., 2018; Wang et al., 2019; Wu & Wang, 2021; Li et al., 2021a; Zheng et al., 2022) have been proposed for purifying backdoor. Techniques such as fine-pruning (Liu et al., 2018) and adversarial neural pruning (Wu & Wang, 2021) require a long training time due to iterative searching criteria. Furthermore, the purification performance deteriorates significantly as the attacks get stronger. In this work, we explore the backdoor insertion and removal phenomena from the DNN optimization point of view. Unlike a benign model, a backdoor model is forced to learn two different data distributions: clean data distribution and poisoned/trigger data distribution. Having to learn both distributions, backdoor model optimization usually leads to a *bad local minima* or sharper minima *w.r.t.* clean distribution. We claim that backdoor can be removed by re-optimizing the model to a smoother minima. One easy re-optimization scheme could be simple DNN weights fine-tuning with a few clean validation samples. However, fine-tuning all DNN parameters often requires huge computational cost and may result in sub-par clean test performance after purification. Therefore, we intend to *fine-tune only one layer* to effectively remove the backdoor.

Fine-tuning only one layer creates a shallow network scenario where SGD-based optimization becomes a bit challenging. Choromanska et al. (2015) claims that the probability of finding bad local minima or poor quality solution increases as the network size decreases. Even though there are

good-quality solutions, it usually requires exponentially long time to find those minima (Choromanska et al., 2015). As a remedy to this, we opt to use a curvature aware optimizer, Natural Gradient Decent (NGD), that has *higher probability of escaping the bad local minima as well as faster convergence rate*, specifically in the shallow network scenario (Amari, 1998; Martens & Grosse, 2015). To this end, we propose a novel backdoor purification technique—Natural Gradient Fine-tuning (NGF)—which focuses on removing backdoor through fine-tuning *only one layer*. However, straightforward application of NGF with simple cross-entropy (CE) loss may result in poor clean test performance. To boost this performance, we use a clean distribution-aware regularizer that prioritizes the update of parameters sensitive to clean data distribution. Our proposed method achieves SOTA performance in a wide range of benchmarks, *e.g.*, four different datasets including *ImageNet*, 13 recent backdoor attacks *etc.* Our contributions can be summarized as follows:

- We analyze the loss surface characteristics of a DNN during backdoor insertion and purification processes. Our analysis shows that the optimization of a backdoor model leads to a *bad local minima* or sharper minima compared to a benign model. We argue that backdoor can be purified by re-optimizing the model to a smoother minima and simple fine-tuning can be a viable way for that. To the best of our knowledge, this is the first work that studies the correlation between loss-surface smoothness and backdoor purification.

- We conduct additional studies on backdoor purification process while fine-tuning different parts of a DNN. We observe that SGD-based one-layer fine-tuning fails to escape bad local minima and a loss surface geometry-aware optimizer can be an easy fix to this.

- We propose a novel backdoor purification technique based on Natural Gradient Fine-tuning (NGF). In addition, we employ a clean distribution-aware regularizer to boost the clean test performance of our proposed method. NGF outperforms recent SOTA methods in a wide range of benchmarks.

## 2 RELATED WORK

**Backdoor Attacks:** Backdoor triggers can exist in the form of dynamic patterns (Li et al., 2020), a single pixel (Tran et al., 2018), sinusoidal strips (Barni et al., 2019), human imperceptible noise (Zhong et al., 2020), natural reflection (Liu et al., 2020), adversarial patterns (Zhang et al., 2021), blending backgrounds (Chen et al., 2017), *etc.* Based on target labels, existing backdoor attacks can generally be classified as poison-label or clean-label backdoor attacks. In poison-label backdoor attack, the target label of the poisoned sample is different from its ground-truth label, *e.g.*, BadNets (Gu et al., 2019), Blended attack (Chen et al., 2017), SIG attack (Barni et al., 2019), WaNet (Nguyen & Tran, 2021), Trojan attack (Liu et al., 2017), and BPPA (Wang et al., 2022). Contrary to the poison-label attack, clean-label backdoor attack doesn't change the label of the poisoned sample (Turner et al., 2018; Huang et al., 2022; Zhao et al., 2020b). Recently, Saha et al. (2022) studied backdoor attacks on self-supervised learning.

**Backdoor Defenses:** Existing backdoor defense methods can be categorized into backdoor detection or purifying techniques. Detection based defenses include trigger synthesis approach (Wang et al., 2019; Qiao et al., 2019; Guo et al., 2020; Shen et al., 2021; Dong et al., 2021; Guo et al., 2021; Xiang et al., 2022; Tao et al., 2022), or malicious samples filtering based techniques (Tran et al., 2018; Gao et al., 2019; Chen et al., 2019). However, these methods only detect the existence of backdoor without removing it. Backdoor purification defenses can be further classified as training time defenses and inference time defenses. Training time defenses include model reconstruction approach (Zhao et al., 2020a; Li et al., 2021b), poison suppression approach (Hong et al., 2020; Du et al., 2019; Borgnia et al., 2021), and pre-processing approaches (Li et al., 2021a; Doan et al., 2020). Although training time defenses are often successful, they suffer from huge computational burden and less practical considering attacks during DNN outsourcing. Inference time defenses are mostly based on pruning approaches such as (Koh & Liang, 2017; Ma & Liu, 2019; Tran et al., 2018; Diakonikolas et al., 2019; Steinhardt et al., 2017). Pruning-based approaches are typically based on model vulnerabilities to backdoor attacks. For example, MCR (Zhao et al., 2020a) and CLP (Zheng et al., 2022) analyzed node connectivity and channel Lipschitz constant to detect backdoor vulnerable neurons. ANP (Wu & Wang, 2021) prune neurons through backdoor sensitivity analysis using adversarial search on the parameter space. Instead, we propose a simple one-layer fine-tuning based defense that is both fast and highly effective. To remove backdoor, our proposed method revisits the DNN fine-tuning paradigm from a novel point of view.

## 3 BACKGROUND

**Attack Model.** We consider an adversary with the capabilities of carrying a backdoor attack on a DNN model, $f_\theta : \mathbb{R}^d \to \mathbb{R}^c$, by training it on a poisoned data set $\mathbb{D}_{\text{train}} = \{X_{\text{train}}, Y_{\text{train}}\}$. Here, $\theta$ is the parameters of the model, $d$ is the input data dimension and $c$ is the total number of classes. The data poisoning happens through specific set of triggers that can only be accessed by the attacker. The adversary goal is to train the model in a way such that any triggered samples $\hat{x} = x + \delta \in \mathbb{R}^d$ will be wrongly misclassified to a target label, $\bar{y}$. Here, $x$ is a clean test sample and $\delta \in \mathbb{R}^d$ represents the trigger pattern with the properties of $||\delta|| \leq \epsilon$; where $\epsilon$ is the trigger magnitude determined by its shape, size and color. We define the *poison rate* as the ratio of poison and clean data in $\mathbb{D}_{\text{train}}$. An attack is considered successful if the model behaves as: $f_\theta(x) = y$ and $f_\theta(\hat{x}) = \bar{y}$, where $y$ is the true label for $x$. We use attack success rate (ASR) for quantifying such success.

**Defense Goal.** We consider a defender with a task to purify the backdoor model $f_\theta$ using a small clean validation set (usually $1 \sim 10\%$ of the training data). The goal is to repair the model in a way such that it becomes immune to attack, *i.e.*, $f_{\theta_p}(\hat{x}) = y$. Here, $f_{\theta_p}$ is the final purified model.

**Natural Gradient Descent (NGD).** Let us consider a model $p(y|x, \theta)$ with parameters $\theta \in \mathbb{R}^N$ to be fitted with input data $\{(x_i, y_i)\}_{i=1}^{|\mathbb{D}_{\text{train}}|}$ from an empirical data distribution $P_{x,y}$, where $x_i \in X_{\text{train}}$ is an input sample and $y_i \in Y_{\text{train}}$ is its label. We try to optimize the model by solving:

$$\theta^* \in \arg\min_\theta \mathcal{L}(\theta), \tag{1}$$

where $\mathcal{L}(\theta) = \mathcal{L}(y, f_\theta(x)) = \mathbb{E}_{(x_i, y_i) \sim P_{x,y}}[-\log p(y|x, \theta)]$ is the expected full-batch cross-entropy (CE) loss. SGD optimizes for $\theta^*$ iteratively following the direction of the steepest descent (estimated by column vector, $\nabla_\theta \mathcal{L}$) and update the model parameters by: $\theta^{(t+1)} \leftarrow \theta^{(t)} - \alpha^{(t)} \cdot \nabla_\theta^{(t)} \mathcal{L}$, where $\alpha$ is the learning rate. Since SGD uses the Identity matrix as the pre-conditioner, it is *uninformed of the geometry of loss surface*.

In NGD, however, the Fisher Information Matrix (FIM) is used as a pre-conditioner, which can be defined as (Martens & Grosse, 2015),

$$F(\theta) = \mathbb{E}_{(x,y) \sim P_{x,y}}[\nabla_\theta \log p(y|x, \theta) \cdot (\nabla_\theta \log p(y|x, \theta))^T] \in \mathbb{R}^{N \times N}. \tag{2}$$

As FIM is a *loss surface curvature matrix*, a careful integration of it in the update rule of $\theta$ will make the optimizer loss surface geometry aware. Such integration leads us to the update equation of NGD, $\theta^{(t+1)} \leftarrow \theta^{(t)} - \alpha^{(t)} \cdot F(\theta^{(t)})^{-1} \nabla_\theta^{(t)} \mathcal{L}$. Here, the natural gradient is defined as $F(\theta^{(t)})^{-1} \nabla_\theta^{(t)} \mathcal{L}$. From the perspective of information geometry, natural gradient defines the *direction in parameter space* which gives largest change in objective **per unit of change in model** ($p(y|x, \theta)$). Per unit of change in model is measured by KL-divergence (Amari, 1998; Park et al., 2000). Note that KL-divergence is well connected with FIM as it can be used as a local quadrature approximation of KL-divergence of *model change*. Eqn. 2 suggests that one requires the knowledge of the original parameter ($\theta$) space to estimate it. Therefore, FIM can be thought of as a mechanism to translate between the geometry of the model ($p(y|x, \theta)$) and the current parameters ($\theta$) of the model. The way natural gradient defined the *direction in parameter space* is contrastive to the stochastic gradient. Stochastic gradient defines the direction in parameter space for largest change in objective **per unit of change in parameter** ($\theta$) measured by Euclidian distance. That is, the gradient direction is solely calculated based on the changes of parameters, without any knowledge of model geometry.

## 4 SMOOTHNESS ANALYSIS OF BACKDOOR MODELS

In this section, we analyze the loss surface geometry of benign, backdoor, and purified models. To study the loss curvature properties of different models, we aim to analyze the Hessian of loss, $H = \nabla_\theta^2 \mathcal{L}$, where we compute $\mathcal{L}$ using the *clean training set*. The Hessian matrix $H$ is symmetric and one can take the spectral decomposition $H = Q\Lambda Q^T$, where $\Lambda = \text{diag}(\lambda_1, \lambda_2, \dots, \lambda_N)$ contains the eigenvalues and $Q = [q_1 q_2 \dots q_N]$ are the eigenvectors of $H$. As a measure for smoothness, we take the maximum eigenvalue, $\lambda_{\text{max}}(= \lambda_1)$, and the trace of the Hessian, $\text{Tr}(H) = \sum_{i=1}^{i=N} \text{diag}(H)_i$. Low values for these two proxies indicate the presence of highly smooth loss surface (Jastrzebski et al., 2020). The Eigen Spectral density plots in Fig. 1a- 1b tell us about the optimization of benign and backdoor models. To create these models, we use the CIFAR10 dataset and train a PreActResNet18 architecture for 200 epochs. To insert the backdoor, we use TrojanNet (Liu et al., 2017) and a poison

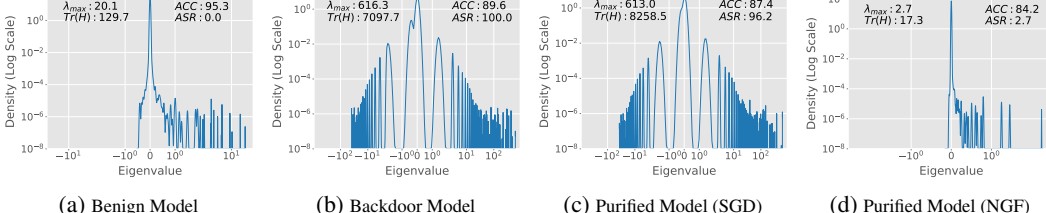

| (a) Benign Model | (b) Backdoor Model | (c) Purified Model (SGD) | (d) Purified Model (NGF) |

Figure 1: Eigen Spectral Density plots of Loss Hessian for (a) benign, (b) backdoor (TrojanNet (Liu et al., 2017)), and (c & d) purified models. In each plot, the maximum eigenvalue ($\lambda_{max}$), trace of Hessian ($\text{Tr}(H)$), clean test accuracy (ACC), and attack success rate (ASR) are also reported. Here, low $\lambda_{max}$ and $\text{Tr}(H)$ hints at the presence of smoother loss surface which often results in low ASR and high ACC. (a & b). Compared to a benign model, a backdoor model tends to reach a sharper minima as shown by the larger range of eigenvalues (x-axis). During purification, SGD optimizer (c) rarely escapes sharp or bad local minima (similar $\lambda_{max}$ and $\text{Tr}(H)$ as the backdoor model) while our proposed method, NGF, (d) converges to a smooth minima. We use CIFAR10 dataset with a PreActResNet18 (He et al., 2016) architecture for all evaluations.

Table 1: Backdoor removal performance while fine-tuning (FT) different parts of a DNN. Fine-tuning only the last layer creates a shallow network scenario. In such scenario, there is a high probability that SGD does not escape bad local minima. Whereas, NGF consistently optimizes to a smooth minima (indicated by low $\lambda_{max}$ for 6 different attacks) which results in backdoor removal, *i.e.*, low ASR and high ACC. We consider CIFAR10 dataset and PreActResNet18 architecture for all evaluations. A clean validation set is used for all purification.

| FT Methods | Badnets $\lambda_{max}$ | ASR | ACC | Blend $\lambda_{max}$ | ASR | ACC | Trojan $\lambda_{max}$ | ASR | ACC | Dynamic $\lambda_{max}$ | ASR | ACC | CLB $\lambda_{max}$ | ASR | ACC | SIG $\lambda_{max}$ | ASR | ACC |
|---|---|---|---|---|---|---|---|---|---|---|---|---|---|---|---|---|---|---|
| Initial | 573.8 | 100 | 92.96 | 715.5 | 100 | 94.11 | 616.3 | 100 | 89.57 | 564.2 | 100 | 92.52 | 717.6 | 100 | 92.78 | 514.1 | 100 | 88.64 |
| Full-Net. | 4.42 | 4.87 | 85.92 | 4.65 | 4.77 | 87.61 | 3.41 | 3.78 | 82.18 | 2.34 | 4.73 | 88.61 | 4.68 | 1.83 | 87.41 | 8.98 | 1.04 | 81.92 |
| CNN-Bbone. | 4.71 | 5.03 | 85.64 | 5.14 | 4.92 | 87.24 | 4.19 | 3.95 | 81.86 | 2.46 | 5.11 | 87.54 | 5.19 | 2.08 | 86.67 | 9.74 | 1.61 | 81.55 |
| Cls. (SGD) | 556.1 | 98.27 | **90.17** | 541.7 | 97.29 | **93.48** | 613.0 | 96.25 | **87.36** | 446.5 | 93.58 | **91.36** | 361.9 | 89.21 | **91.73** | 563.2 | 96.70 | **86.92** |
| Cls. (NGF) | **2.79** | **1.86** | 88.32 | **2.43** | **0.38** | 91.17 | **2.74** | **2.64** | 84.21 | **1.19** | **1.17** | 90.97 | **3.13** | **1.04** | 88.37 | **1.48** | **0.12** | 84.16 |

rate of 10%. From the comparison of $\lambda_{max}$ and $\text{Tr}(H)$, we can conjecture that optimization of a benign model produces smoother loss surface. We observe similar phenomena for different datasets and architectures; details are in *Appendix F*. The main difference between a benign and a backdoor model is that the latter needs to learn two different data distributions: clean and poison. Based on our observations, we state following conjectures:

**Conjecture 1.** Having to learn two different data distributions, a backdoor model reaches a sharper minima, *i.e.*, large $\lambda_{max}$ and $\text{Tr}(H)$, as compared to the benign model.

We support this conjecture with empirical evidence presented in Table 1. Looking at the $\lambda_{max}$ in the 'Initial' row for all 6 attacks (details are in *Appendix D*), it can be observed that all of these backdoor models optimizes to a sharp minima. As these models are optimized on both distributions, they also have high attack success rates (ASR) as well as high clean test accuracy (ACC). Note that, the measure of smoothness is done *w.r.t.* clean data distribution. The use of clean distribution in our smoothness analysis is driven from the practical consideration as our particular interest lies with the performance *w.r.t.* clean distribution; more details are in *Appendix C.1*. Since high ASR and ACC indicate that the model had learned both distributions, it supports Conjecture 1.

**Conjecture 2.** Through *proper* fine-tuning with clean validation data, a backdoor model can be re-optimized to a smoother minima *w.r.t.* clean data distribution. Optimization to a smoother minima leads to backdoor purification, *i.e.*, low ASR and high ACC.

By *proper fine-tuning*, we imply that the fine-tuning will lead to an optimal solution *w.r.t.* the data distribution we fine-tune the model with. To support Conjecture 2, we show the removal performances of fine-tuning based purification methods in Table 1. To remove backdoor using a clean validation set (~1% of train-set), we fine-tune different parts of the DNN for 100 epochs with a learning rate of 0.01. As shown in Table 1, after proper fine-tuning (Full-Net, CNN-Bbone), the backdoor model re-optimizes to a smoother minima that leads to successful backdoor removal.

**One-Layer Fine-tuning:** We observe that one can remove the backdoor by fine-tuning either the full network or only the CNN backbone (using SGD). However, these methods can be computationally costly and less practical. Furthermore, such fine-tuning often leads to high drop in ACC. As an alternative, one could fine-tune only the last or classification (Cls.) layer. However, even with a small validation set, a one-layer network becomes a shallow network to optimize. According to the

spin-glass analogy in Choromanska et al. (2015), as the network size decreases the probability for the SGD optimizer to find *sharp local minima or poor quality minima* increases accordingly. In case of shallow network, the quality of minima is decided by their distances from the global minima. Choromanska et al. (2015) also observes that the process of finding a path from bad local minima to a good quality solution or global minima takes *exponentially long time*. Therefore, it is not always feasible to use the SGD optimizer for shallow network. Table 1 (row–Cls. (SGD)) corroborates this hypothesis as SGD optimizer fails to escape the sharp minima resulting in similar ASRs as the initial backdoor model. Instead of using SGD, one can use natural gradient descent (NGD) that has *higher probability of escaping the bad local minima as well as faster convergence rate*, specifically in the shallow network scenario (Amari, 1998; Martens & Grosse, 2015). Therefore, to effectively purify a backdoor model, we propose a novel Fisher Information matrix based backdoor purification objective function and optimize it using the NGD optimizer.

## 4.1 NATURAL GRADIENT FINE-TUNING (NGF)

Let us decompose the model parameters $\theta$ as,

$$\theta = \{\mathbf{W}_{0,1}, \mathbf{W}_{1,2}, \mathbf{W}_{2,3}, \cdots, \mathbf{W}_{L-1,L}\}$$

here, $\mathbf{W}_{i,i+1}$ is the parameters between layer $i$ and layer $i+1$, commonly termed as $(i+1)^{th}$ layer's parameters. $\mathbf{W}_{L-1,L}$ is the $L^{th}$ layer's (Cls. layer) parameters and we are particularly interested in fine-tuning only this layer. Now, consider a validation set, $\mathbb{D}_{\mathsf{val}} = \{X_{\mathsf{val}}, Y_{\mathsf{val}}\}$ that contains only clean samples. We denote $\theta_L (= \mathbf{W}_{L-1,L})$ as the $L^{th}$ layer's parameters. To purify the backdoor model, we formulate the following loss

$$\mathcal{L}_p(y, f_\theta(x)) \approx \mathcal{L}(y, f_\theta(x)) + \frac{\eta}{2} \sum_i \mathsf{diag}(F(\bar{\theta}_L))_i \cdot (\theta_{L,i} - \bar{\theta}_{L,i})^2, \quad (3)$$

which is a combination of the CE loss on the validation set and a regularizer. Here, $\bar{\theta}_L$ (fixed) is $L^{th}$ layer parameters of the initial backdoor model, *i.e.*, $\theta_L^{(0)} = \bar{\theta}_L$ .

In a backdoor model, some neurons/parameters are more vulnerable than others. The vulnerable parameters are believed to be the ones that are sensitive to poison/trigger data distribution (Wu & Wang, 2021). In general, CE loss does not discriminate whether a parameter is more sensitive to clean or poison distribution. Such lack of discrimination may allow drastic/unwanted changes to the parameters responsible for learned clean distribution. This usually leads to sub-par clean test accuracy after purification and it requires additional measures to fix this issue. Motivated by Kirkpatrick et al. (2017), we introduce a *clean distribution aware regularization* term as a product of two terms: i) an error term that accounts for the deviation of $\theta_L$ from $\bar{\theta}_L$; ii) a vector, $\mathsf{diag}(F(\bar{\theta}_L))$, consisting of the diagonal elements of FIM ($F(\bar{\theta}_L)$). As the first term controls the changes of parameters *w.r.t.* $\bar{\theta}_L$, it helps the model to remember the already learned distribution. However, learned data distribution consists of clean and poison distribution both. To explicitly force the model to remember the *clean distribution*, we compute $F(\bar{\theta}_L)$ using a *clean* validation set; with similar distribution as the learned clean data. Note that, $\mathsf{diag}(F(\bar{\theta}_L))_i$ represents the square of the derivative of log-likelihood of clean distribution *w.r.t.* $\bar{\theta}_{L,i}$, $[\nabla_{\bar{\theta}_{L,i}} \log p(y|x, \theta)]^2$ (ref. eqn. (6)). In other words, $\mathsf{diag}(F(\bar{\theta}_L))_i$ is the measure of importance of $\bar{\theta}_{L,i}$ towards remembering the learned clean distribution. If $\mathsf{diag}(F(\bar{\theta}_L))_i$ has a higher importance, we allow minimal changes to $\bar{\theta}_{L,i}$ over the purification process. This careful design of such regularizer improves the clean test performance significantly. We use $\eta$ as a regularization constant.

The overall optimization problem using the loss-function defined in (3) for purifying the backdoor model $f_\theta$ is as follows:

$$\text{Objective function:} \quad \theta_p := \underset{\theta_L}{\arg\min}\, \mathcal{L}_p(y, f_\theta(x)); \ \ x \in X_{val}, \ y \in Y_{val} \quad (4)$$

$$\text{Update Policy:} \quad \theta_L^{(t+1)} \leftarrow \theta_L^{(t)} - \alpha F(\theta_L^{(t)})^{-1} \nabla_{\theta_L} \mathcal{L}_p \quad (5)$$

$$\text{where,}\ F(\theta_L) := \frac{1}{n} \sum_{j=1}^{n} \left( \nabla_{\theta_L} \log p(y_j|x_j, \theta) \cdot (\nabla_{\theta_L} \log (y_j|x_j, \theta))^T \right). \quad (6)$$

Here, $F \in \mathbb{R}^{|\theta_L| \times |\theta_L|}$ is the FIM, and $n$ is the validation set size. Notice that, as we only consider fine-tuning of $L^{th}$-layer, the computation of $F$ and $F^{-1}$ ($|\theta_L| \times |\theta_L|$ matrices) becomes tractable.

Table 2: Comparison of different defense methods for four benchmark datasets. Backdoor removal performance, *i.e.*, drop in ASR, against a wide range of attacking strategies show the effectiveness of NGF. For CIFAR10 and GTSRB, the poison rate is 10%. For Tiny-ImageNet and ImageNet, we employ ResNet34 and ResNet50 architectures, respectively. We use a poison rate of 5% for these 2 datasets and report performance on successful attacks (ASR close to 100%) only. Average drop (↓) indicates the % changes in ASR/ACC compared to the baseline, *i.e.*, ASR/ACC of *No Defense*. Higher ASR drop and lower ACC drop is desired for a good defense.

| Dataset | Method | No Defense | | Vanilla FT | | ANP | | I-BAU | | AWM | | NGF (Ours) | |
|---|---|---|---|---|---|---|---|---|---|---|---|---|---|
| | Attacks | ASR | ACC | ASR | ACC | ASR | ACC | ASR | ACC | ASR | ACC | ASR | ACC |
| CIFAR-10 | *Benign* | 0 | 95.21 | 0 | 92.28 | 0 | 93.98 | 0 | 93.56 | 0 | 93.80 | 0 | **94.10** |
| | Badnets | 100 | 92.96 | 4.87 | 85.92 | 2.84 | 85.96 | 9.72 | 87.85 | 4.34 | 86.17 | 1.86 | **88.32** |
| | Blend | 100 | 94.11 | 4.77 | 87.61 | 3.81 | 89.10 | 11.53 | 90.84 | 2.13 | 88.93 | 0.38 | **91.17** |
| | Troj-one | 100 | 89.57 | 3.78 | 82.18 | 5.47 | 85.20 | 7.91 | **87.24** | 5.41 | 86.45 | 2.64 | 84.21 |
| | Troj-all | 100 | 88.33 | 3.91 | 81.95 | 5.53 | 84.89 | 9.82 | 85.94 | 4.42 | 84.60 | 2.79 | **86.10** |
| | SIG | 100 | 88.64 | 1.04 | 81.92 | 0.37 | 83.60 | 4.12 | 83.57 | 0.90 | 83.38 | 0.12 | **84.16** |
| | Dyn-one | 100 | 92.52 | 4.73 | 88.61 | 1.78 | 86.26 | 10.48 | 89.16 | 3.35 | 88.41 | 1.17 | **90.97** |
| | Dyn-all | 100 | 92.61 | 4.28 | 88.32 | 2.19 | 84.51 | 10.30 | 89.74 | 2.46 | 87.72 | 1.61 | **90.19** |
| | CLB | 100 | 92.78 | 1.83 | 87.41 | 1.41 | 85.07 | 5.78 | 86.70 | 1.89 | 84.18 | 1.04 | **88.37** |
| | CBA | 93.20 | 90.17 | 27.80 | 83.79 | 45.11 | 85.63 | 36.12 | 85.05 | 38.81 | 85.58 | 24.60 | **85.97** |
| | FBA | 100 | 90.78 | 7.95 | 82.90 | 66.70 | **87.42** | 10.66 | 87.35 | 22.31 | 87.06 | 6.21 | 86.96 |
| | WaNet | 98.64 | 92.29 | 5.81 | 86.70 | 3.18 | 89.24 | 10.72 | 85.94 | 2.96 | 89.45 | 2.38 | **89.65** |
| | ISSBA | 99.80 | 92.80 | 6.76 | 85.42 | **3.82** | 89.20 | 12.48 | 90.03 | 4.57 | 89.59 | 4.24 | **90.18** |
| | BPPA | 99.70 | 93.82 | 9.94 | 90.23 | 10.46 | 90.57 | 9.94 | 90.68 | 10.60 | 90.88 | 7.14 | **91.84** |
| | Avg. Drop | - | - | 92.61↓ | 6.03↓ | 87.59↓ | 4.98↓ | 87.82↓ | 3.95↓ | 91.32↓ | 4.53↓ | **95.01↓** | **3.33↓** |
| GTSRB | *Benign* | 0 | 97.87 | 0 | 93.08 | 0 | 95.42 | 0 | 96.18 | 0 | 95.32 | 0 | **95.76** |
| | Badnets | 100 | 97.38 | 1.36 | 88.16 | 0.35 | 93.17 | 2.72 | **94.55** | 2.84 | 93.58 | 0.24 | 94.11 |
| | Blend | 100 | 95.92 | 5.08 | 89.32 | 4.41 | 93.02 | 4.13 | **94.30** | 4.96 | 92.75 | 2.91 | 93.31 |
| | Troj-one | 99.50 | 96.27 | 2.07 | 90.45 | 1.81 | 92.74 | 3.04 | 93.17 | 2.27 | 93.56 | 1.21 | **94.18** |
| | Troj-all | 99.71 | 96.08 | 2.48 | 89.73 | 2.16 | 92.51 | 2.79 | 93.28 | 1.94 | 92.84 | 1.58 | **93.87** |
| | SIG | 97.13 | 96.93 | **1.93** | 91.41 | 6.17 | 91.82 | 2.64 | 93.10 | 5.32 | 92.68 | 3.24 | **93.48** |
| | Dyn-one | 100 | 97.27 | 2.27 | 91.26 | 2.08 | 93.15 | 5.82 | **95.54** | 1.89 | 93.52 | 1.51 | 94.27 |
| | Dyn-all | 100 | 97.05 | 2.84 | 91.42 | 2.49 | 92.89 | 4.87 | 93.98 | 2.74 | 93.17 | 1.26 | **94.14** |
| | BPPA | 99.18 | 98.12 | 5.14 | 94.48 | 7.19 | 93.79 | 8.63 | 94.50 | 5.43 | 94.22 | 4.45 | **95.27** |
| | Avg. Drop | - | - | 96.54↓ | 6.10↓ | 96.10↓ | 3.99↓ | 95.11↓ | 2.83↓ | 96.02↓ | 3.59↓ | **97.39↓** | **2.79↓** |
| Tiny-ImageNet | *Benign* | 0 | 62.56 | 0 | 58.20 | 0 | 59.29 | 0 | 59.34 | 0 | 59.08 | 0 | **59.67** |
| | Badnets | 100 | 59.80 | 3.84 | 53.58 | 61.23 | 55.41 | 13.29 | 54.56 | 31.44 | 54.81 | 2.34 | **55.84** |
| | Trojan | 100 | 59.16 | 6.77 | 52.62 | 79.56 | 54.76 | 11.94 | **55.10** | 38.23 | 54.28 | 3.38 | 54.87 |
| | Blend | 100 | 60.11 | 2.18 | 51.22 | 81.58 | 54.70 | 17.42 | 54.19 | 41.37 | 53.78 | 1.58 | **54.98** |
| | SIG | 98.48 | 60.01 | 5.02 | 52.18 | 28.67 | 54.71 | 9.31 | **55.72** | 27.68 | 54.11 | 2.81 | 54.63 |
| | CLB | 97.71 | 60.33 | 5.61 | 51.68 | 16.24 | 55.18 | 10.68 | 54.93 | 36.52 | 55.02 | 4.06 | **55.40** |
| | Avg. Drop | - | - | 94.55↓ | 7.63↓ | 45.38↓ | 4.93↓ | 86.71↓ | 4.98↓ | 64.19↓ | 5.48↓ | **96.40↓** | **4.74↓** |
| ImageNet | *Benign* | 0 | 77.06 | 0 | 73.52 | 0 | 68.85 | 0 | 74.21 | 0 | 71.63 | 0 | **74.51** |
| | Badnets | 99.24 | 74.53 | 5.91 | 69.37 | 43.31 | 66.28 | 21.87 | 69.46 | 21.18 | 69.44 | 4.61 | **70.46** |
| | Trojan | 99.21 | 74.02 | 4.63 | 69.15 | 38.81 | 66.14 | 25.74 | 69.35 | 28.85 | 68.62 | 4.02 | **69.97** |
| | Blend | 100 | 74.42 | 4.43 | 70.20 | 57.79 | 65.51 | 27.45 | 68.61 | 34.15 | 68.91 | 3.83 | **70.52** |
| | SIG | 94.66 | 74.69 | 3.23 | 69.82 | 16.28 | 66.08 | 15.37 | 70.02 | 16.47 | 69.74 | 2.94 | **71.36** |
| | CLB | 95.06 | 74.14 | 3.71 | 69.19 | 18.37 | 66.41 | 21.64 | 69.70 | 23.50 | 69.32 | 3.05 | **70.25** |
| | Avg. Drop | - | - | 93.25↓ | 4.81↓ | 62.72↓ | 8.28↓ | 75.22↓ | 4.93↓ | 72.80↓ | 5.15↓ | **93.94↓** | **3.85↓** |

After solving the above optimization problem, we will get modified parameters, $\overline{\mathbf{W}}_{L-1,L}$. Finally, we get the purified model, $f_{\theta_p}$ with $\theta_p$ as

$$\theta_p = \{\mathbf{W}_{0,1}, \mathbf{W}_{1,2}, \mathbf{W}_{2,3}, \cdots, \overline{\mathbf{W}}_{L-1,L}\}$$

Fig. 1c-1d show that NGF indeed does reach the smooth minima as opposed to SGD based fine-tuning. We provide additional results in Table 1 for both NGF and SGD. Notice that the purified model seems to have a smoother loss surface than the benign model (2.7 vs. 20.1 for $\lambda_{\max}$). This, however, does not translate to better ACC than the benign model. The ACC of the purified model is always bounded by the ACC of the backdoor model. To the best of our knowledge, our study on the correlation between loss-surface smoothness and backdoor purification is novel. NGF is also the first method to employ a second-order optimizer for purifying backdoor. *More details are in* Appendix C

## 5 EXPERIMENTAL RESULTS

### 5.1 EVALUATION SETTINGS

**Datasets:** To begin with, we evaluate our proposed method through conducting a wide range of experiments on two widely used datasets for backdoor attack study: **CIFAR10** (Krizhevsky et al., 2009) with 10 classes, **GTSRB** (Stallkamp et al., 2011) with 43 classes. As a test of scalability, we also consider **Tiny-ImageNet** (Le & Yang, 2015) with 100,000 images distributed among 200 classes and **ImageNet** (Deng et al., 2009) with 1.28M images distributed among 1000 classes.

**Attacks Configurations:** We consider 13 state-of-the-art backdoor attacks: 1) Badnets (Gu et al., 2019), 2) Blend attack (Chen et al., 2017), 3 & 4) TrojanNet (Troj-one & Troj-all) (Liu et al., 2017), 5) Sinusoidal signal attack (SIG) (Barni et al., 2019), 6 & 7) Input-Aware Attack (Dyn-one and Dyn-all) (Nguyen & Tran, 2020), 8) Clean-label attack (CLB) (Turner et al., 2018), 9) Composite

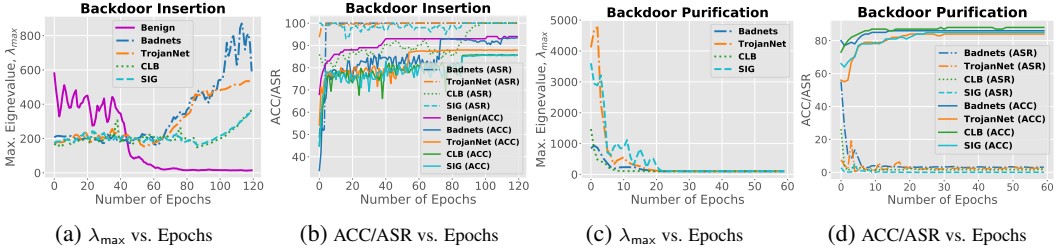

(a) $\lambda_{max}$ vs. Epochs    (b) ACC/ASR vs. Epochs    (c) $\lambda_{max}$ vs. Epochs    (d) ACC/ASR vs. Epochs

Figure 2: Loss Surface characteristics of a DNN during backdoor insertion and purification processes. a & b) As the joint optimization on clean and poison distribution progresses, *i.e.*, high ACC & ASR, the loss surface becomes less and less smoother, *i.e.*, high $\lambda_{max}$). c & d) One can purify backdoor by gradually making the loss surface smoother. We use CIFAR10 dataset with four different attacks.

backdoor (CBA) (Lin et al., 2020), 10) Deep feature space attack (FBA) (Cheng et al., 2021), 11) Warping-based backdoor attack (WaNet) (Nguyen & Tran, 2021), 12) Invisible triggers based backdoor attack (ISSBA) (Li et al., 2021c), and 13) Quantization and contrastive learning based attack (BPPA) (Wang et al., 2022). To ensure fair comparison, we follow the similar trigger patterns and settings as in their original papers. In Troj-one and Dyn-one attacks, all of the triggered images have same target label. On the other hand, target labels are uniformly distributed over all classes for Troj-all and Dyn-all attacks. For creating these attacks on CIFAR10 and GTSRB, we use a poison rate of 10% and train a PreActResNet18 (He et al., 2016) and a WideResNet-16-1 (Zagoruyko & Komodakis, 2016) architectures, respectively, for 250 epochs with an initial learning rate of 0.01. More details on hyper-parameters and overall training settings can be found in *Appendix D*.

**Defenses Configurations:** We compare our approach with 4 existing backdoor mitigation methods: 1) Vanilla Fine-Tuning (FT); where we fine-tune all DNN parameters, 2) Adversarial Neural Pruning (ANP) (Wu & Wang, 2021) with 1% clean validation data, 3) Implicit Backdoor Adversarial Unlearning (I-BAU) (Zeng et al., 2021) 4) Adversarial Weight Masking (AWM) (Chai & Chen, 2022). We also compare NGF with another recent defense technique described in (Zheng et al., 2022). However, we present this comparison in the *Appendix E* due to several performance issues.[1] To apply NGF on CIFAR10, we fine-tune the last layer of the DNN for $E_p$ epochs with 1% clean validation data. Here, $E_p$ is the number of purification epochs and we choose a value of 100 for this. For optimization, we choose a learning rate of 0.01 with a decay rate of 0.1/40 epochs and consider regularization constant $\eta$ to be 0.1. Additional experimental details for NGF and other defense methods are in *Appendix D.3*. For GTSRB, we increase the validation size to 3% as there are less samples available per class. Rest of the training settings are same as CIFAR10. For NGF on Tiny-ImageNet, we consider a validation size of 5% as a size less than this seems to hurt clean test performance (after purification). We fine-tune the model for 15 epochs with an initial learning rate of 0.01 with a decay rate of 0.3/epoch. Finally, we validate the effectiveness of NGF on ImageNet. For removing the backdoor, we use 3% validation data and fine-tune for 2 epochs. A learning rate of 0.001 has been employed with a decay rate of 0.005 per epoch. *We define the effectiveness of a defense method in terms of average drop in ASR and ACC over all attacks. A highly effective method should have a high drop in ASR with a low drop in ACC.* We define ASR as the percentage of poison test samples that are classified to the adversary-set target label.

### 5.2 PERFORMANCE EVALUATION OF NGF

In Table 2, we present the performance of different defenses for four different datasets.

**CIFAR10:** We consider five *label poisoning attacks*: Badnets, Blend, TrojanNet, Dynamic, and BPPA. For TorjanNet, we consider two different variations based on label-mapping criteria: Troj-one and Troj-all. Regardless the complexity of the label-mapping type, our proposed method outperforms all other methods both in terms of ASR and ACC. We also create two variations for Dynamic attack: Dyn-one and Dyn-all. Dynamic attack optimizes for input-aware triggers that are capable of fooling the model; making it more challenging than the static trigger based attacks (Badnets, Blend and Trojan). However, NGF outperforms other methods by a satisfactory margin. We also consider attacks that does not change the label during trigger insertion, *i.e.*, *clean label attack*. Two such attacks are CLB and SIG. For further validation of our proposed method, we use *deep feature based attacks*, CBA and FBA. Both of these attacks manipulates deep features for backdoor insertion. Compared

---

[1]Based on our re-run, we notice significantly larger drop in ACC as compared to other defenses.

to other defenses, NGF shows better effectiveness against these diverse set of attacks achieving an average drop of 95.01% in ASR while sacrificing an ACC of 3.33% for that. Table 2 also shows the performance of baseline methods such as I-BAU and AWM. AWM performs similarly as ANP and often struggles to remove the backdoor.

**GTSRB:** In case of GTSRB, almost all defenses perform similarly for Badnets and Trojan. This, however, does not hold for blend as we achieve an 2.17% ASR improvement over the next best method. The performance is consistent for other attacks as well. Overall, we record an average 97.39% ASR drop with only an 2.79% drop in ACC. *In some cases, ACC for I-BAU are slightly better as it uses a much larger validation size (5%) for purification than other defense techniques.*

**ImageNet:** For scalability test of NGF, we consider two large and widely used datasets, Tiny-ImageNet and ImageNet. In consistence with other datasets, NGF obtains SOTA performance in these diverse datasets too. The effectiveness of ANP reduces significantly for this dataset. In case of large models and datasets, the task of identifying and pruning vulnerable neurons gets more complicated and may result in wrong neurons pruning.

### 5.3 ABLATION STUDIES

**Smoothness Analysis of Different Attacks:** We show the relationship between loss surface smoothness and backdoor insertion process in Fig. 2a-2b. During backdoor insertion, the model is optimized for 2 different data distributions: clean and poison. Compared to a benign model, the loss surface of a backdoor *becomes much sharper as the model becomes well optimized for both distributions*, *i.e.*, model has both high ASR and high ACC. At the beginning of training, both backdoor and benign models are far from being well optimized. The difference between these models are prominent once the model reaches closer to the final optimization point. As shown in Fig. 2b, the training becomes reasonably stable after 100 epochs with ASR and ACC near saturation level. Comparing $\lambda_{\max}$ of benign and all backdoor models after 100 epochs, we notice a sharp contrast in Fig. 2a. This validates our previous claim on loss surface smoothness of benign and backdoor models.

During purification period as shown in 2c-2d, the model is being optimized to a smoother minima. As a result, ASR becomes close to 0 while retaining good clean test performance. Note that, we calculate loss Hessian and $\lambda_{\max}$ using all DNN parameters.

Table 3: Performance comparison of NGF to other SGD-based optimizers. A more suitable sharpness-aware SGD-based optimizer is also considered here. However, NGF is far more effective in purifying backdoor (lower ASR) due to its consistent convergence to smooth minima. We use CIFAR10 dataset for these evaluations.

| Defense | No Defense | | AdaGrad | | RMSProp | | Adam | | SAM | | NGF (Ours) | |
|---|---|---|---|---|---|---|---|---|---|---|---|---|
| Attacks | ASR | ACC | ASR | ACC | ASR | ACC | ASR | ACC | ASR | ACC | ASR | ACC |
| Badnets | 100 | 92.96 | 96.54 | 91.16 | 98.33 | **91.73** | 97.68 | 91.45 | 91.08 | 90.12 | **1.86** | 88.32 |
| Blend | 100 | 94.11 | 97.43 | 91.67 | 95.41 | **92.21** | 94.79 | 92.15 | 89.25 | 91.11 | **0.38** | 91.17 |
| Trojan | 100 | 89.57 | 95.52 | **88.51** | 94.87 | 88.02 | 96.74 | 87.98 | 92.15 | 88.33 | **2.64** | 84.21 |
| Dynamic | 100 | 92.52 | 97.37 | **91.45** | 93.50 | 91.12 | 96.90 | 91.40 | 92.24 | 90.79 | **1.17** | 90.97 |
| SIG | 100 | 88.64 | 86.20 | 87.98 | 86.31 | 87.74 | 85.66 | 87.75 | 81.68 | **88.04** | **0.31** | 83.14 |
| CLB | 100 | 92.78 | 96.81 | 90.86 | 95.53 | 90.96 | 95.87 | **91.02** | 91.04 | 90.97 | **1.04** | 88.37 |

ters. This indicates that changing the parameters of only one layer impacts the loss landscape of whole network. Even though the CNN-backbone parameters are frozen, NGF changes the last layer in a way such that whole backdoor network behaves differently, *i.e.*, like a benign model.

**Evaluation of Different Optimizers:** We compare the performance of NGF with different variants of first-order optimizer: (i) *AdaGrad* (Duchi et al., 2011), (ii) *RMSProp* (Hinton et al.), (iii) *Adam* (Kingma & Ba, 2014), and (iv) Sharpness-Aware Minimization (*SAM*) (Foret et al., 2020) is a recently proposed SGD-based optimizer that explicitly penalizes the abrupt changes of loss surface by bounding the search space within a small region. This forces the changes of model parameters in a way such that the optimization achieve smoother loss surface. Table 3 shows that NGF outperforms all of these variants of first-order optimizer by a huge margin. At the same time, proposed method achieves comparable clean test performance. Although SAM usually performs better than vanilla SGD in terms of smooth DNN optimization, SAM's performance in shallow network scenario (our case) is almost similar to vanilla SGD. Two potential reasons behind this poor performance are (i) using a predefined local area to search for maximum

Table 4: Avg. runtime comparison for different datasets. Here, #Parameters is the total number of parameters in the last layer. An NVIDIA RTX 3090 GPU is used for all experiments.

| Dataset | # Parameters | Method | Runtime (Sec.) |
|---|---|---|---|
| CIFAR10 | 5120 | FT | 78.1 |
| | | NGF | **38.3** |
| GTSRB | 22016 | FT | 96.2 |
| | | NGF | **47.4** |
| Tiny-ImageNet | 409.6K | FT | 637.6 |
| | | NGF | **374.2** |
| ImageNet | 2.048M | FT | 2771.6 |
| | | NGF | **1681.4** |

Table 5: Performance of SGD-Long and NGF while fine-tuning only the last layer of DNN. For SGD-Long, we consider a long purification period with $E_p = 2500$. NGF performance with and without the regularization term underlines the importance of the proposed regularizer. The results shown here are for CIFAR10 dataset.

| Methods | Badnets | | Blend | | Trojan | | Dynamic | | CLB | | SIG | | CBA | | Runtime |
|---|---|---|---|---|---|---|---|---|---|---|---|---|---|---|---|
| | ASR | ACC | ASR | ACC | ASR | ACC | ASR | ACC | ASR | ACC | ASR | ACC | ASR | ACC | (Secs.) |
| Initial | 100 | 92.96 | 100 | 94.11 | 100 | 89.57 | 100 | 92.52 | 100 | 92.78 | 100 | 88.64 | 93.20 | 90.17 | – |
| SGD-Long | 82.34 | **90.68** | 7.13 | **92.46** | 86.18 | **87.29** | 57.13 | 90.51 | 13.84 | 88.11 | 0.26 | **85.74** | 84.41 | **86.87** | 907.5 |
| NGF w/o Reg. | 1.91 | 87.65 | **0.31** | 90.54 | 3.04 | 83.31 | 1.28 | 90.24 | **0.92** | 87.13 | 0.16 | 84.46 | 25.58 | 84.81 | **37.8** |
| NGF | **1.86** | 88.32 | 0.38 | 91.17 | **2.64** | 84.21 | **1.17** | **90.97** | 1.04 | **88.37** | **0.12** | 84.16 | **24.60** | 85.97 | 38.3 |

Table 6: Evaluation of NGF on backdoor attacks with high poison rates, upto 50%. We consider CIFAR10 dataset and two closely performing defenses for this comparison.

| Attack | BadNets | | | | | | Blend | | | | | | Trojan | | | | | |
|---|---|---|---|---|---|---|---|---|---|---|---|---|---|---|---|---|---|---|
| Poison Rate | 25% | | 35% | | 50% | | 25% | | 35% | | 50% | | 25% | | 35% | | 50% | |
| Method | ASR | ACC | ASR | ACC | ASR | ACC | ASR | ACC | ASR | ACC | ASR | ACC | ASR | ACC | ASR | ACC | ASR | ACC |
| *No Defense* | 100 | 88.26 | 100 | 87.43 | 100 | 85.11 | 100 | 86.21 | 100 | 85.32 | 100 | 83.28 | 100 | 87.88 | 100 | 86.81 | 100 | 85.97 |
| ANP | 7.81 | 82.22 | 16.35 | 80.72 | 29.80 | 78.27 | 29.96 | **82.84** | 47.02 | 78.34 | 86.29 | 69.15 | 11.96 | 76.28 | 63.99 | 72.10 | 89.83 | 70.02 |
| FT | 5.21 | 78.11 | 8.39 | 74.06 | 11.52 | 69.81 | 1.41 | 68.73 | 4.56 | 63.87 | 7.97 | 55.70 | 3.98 | 76.99 | 4.71 | 72.05 | 5.59 | 70.98 |
| **NGF (Ours)** | **2.12** | **85.50** | **2.47** | **84.88** | **4.53** | **82.32** | **0.83** | 80.62 | **1.64** | **79.62** | **2.21** | **76.37** | **3.02** | **83.10** | **3.65** | **81.66** | **4.66** | **80.30** |

loss, and (ii) using 'Euclidean distance' metric instead of geometric distance metric. In contrast, NGD with curvature geometry aware Fisher Information Matrix can successfully avoid such bad minima and optimizes to a global minima.

**Runtime Analysis:** In Table 4, we show the average runtime for different defenses. Similar to purification performance, purification time is also an important indicator to measure the success of a defense technique. In Section 5.2, we already show that our method outperforms other defenses in most of the settings. As for the run time, our method completes the purification (for CIFAR10) in just 38.3 seconds; which is almost half as compared to FT. The time-advantage of our method also holds for large datasets and models, *e.g.*, ImageNet and ResNet50. Runtime comparison with other defenses is in the *Appendix H*.

**Effect of Proposed Regularizer:** In this section, we analyze the effect of regularizer and long training with SGD. The effect of our clean distribution-aware regularizer can be observed in Table 5. NGF with the proposed regularizer achieves an 1% clean test performance improvement over vanilla NGF. For long training with SGD (SGD-Long), we fine-tune the last layer for 2500 epochs. Table 5 shows the evaluations of SGD-Long on 7 different attacks. Even though the ASR performance improves significantly for CLB and SIG attacks, SGD-based FT still severely underperforms for other attacks. Moreover, the computational time increases significantly over NGF. Thus, our choice of *NGD-based FT as a fast and effective backdoor purification technique* is well justified.

**Strong Backdoor Attacks:** By increasing the poison rates, we create stronger version of different attacks against which most defense techniques fail quite often. We use 3 different poison rates, $\{25\%, 35\%, 50\%\}$. We show in Table 6 that NGF is capable of defending very well even with a poison rate of 50%, achieving a significant ASR improvement over FT. Furthermore, there is a sharp difference in classification accuracy between NGF and other defenses. For 25% Blend attack, however, ANP offers a slightly better performance than our method. However, ANP performs poorly in terms of removing backdoor as it obtains an ASR of 29.96% as compared to 0.83% for NGF.

## 6 CONCLUSION

We propose a novel backdoor purification technique based on natural gradient descent fine-tuning. The proposed method is motivated by our analysis of loss surface smoothness and its strong correlation with the backdoor insertion and purification processes. As a backdoor model has to learn an additional data distribution, it tends to be optimized to bad local minima or sharper minima compared to a benign model. We argue that backdoor can be removed by re-optimizing the model to a smoother minima. We further argue that fine-tuning a single layer is enough to remove the backdoor. Therefore, in order to achieve a smooth minima in a single-layer fine-tuning scenario, we propose using an FIM-based DNN objective function and minimize it using a curvature-aware NGD optimizer. Our proposed method achieves SOTA performance in a wide range of benchmarks. Since we fine-tune only one layer the training time overhead reduces significantly, making our method one of the fastest among SOTA defenses. In the future, we aim to extend our smoothness analysis to 3D point-cloud attacks as well as attacks on contrastive learning.

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

## A   APPENDIX

Section B describes our proposed algorithm. Section C discusses the intuitions behind the uses of clean data distribution for smoothness analysis, more explanation on why backdoor behavior and weight loss-landscape are related and smoothness helps to mitigate the effect of backdoor, how FIM helps to smooth the loss-landscape, and why SAM (Foret et al., 2021) does not work in proposed problem setup. Section D contains the experimental details of different attacks and defenses. Section E contain comparison with another recent defense technique. We present smoothness analysis with different datasets and architectures in Section F. Section G shows the *Label Correction Rates* for different defense techniques. Section H and I contain additional runtime analysis. Section K contains more ablation studies purification with last layer re-initialization. Section L and M discuss more attacks, all2all and combined attacks. Our code is available at anonymous *GitHub link* [2].

## B   NATURAL GRADIENT FINE-TUNING

In our proposed method, we aim to remove backdoor by fine-tuning only the last layer. The manner in which we perform that fine-tuning is described in Algorithm 1. After purification, the model should behave like a benign/clean model producing same prediction irrespective of the presence of trigger. Note that Kirkpatrick et al. (2017) proposed a similar regularizer, known as elastic weight consolidation (EWC) used in continual learning. EWC helps a model to learn a new task while keeping the knowledge of previously learned tasks. Our clean data distribution aware regularization, instead, only helps to preserve the knowledge of previously learned task corresponding to clean data distribution (not task corresponding to poison samples) through computing $\bar{F}$ using clean validation set. Thus, the design of our regularizer is backdoor specific and needs careful attention as it is crucial for clean test performance.

---

**Algorithm 1:** Natural Gradient Fine-tuning

**Input:** Backdoor Model ($f_\theta(.)$), 1% Clean Validation Set $\mathbb{D}_{val}$, Number of Purification Epochs $\mathcal{N}$
$\mathcal{X}, \mathcal{Y} \leftarrow \mathbb{D}_{val}$
$\bar{F} \leftarrow \frac{1}{|\mathbb{D}_{val}|} \sum_{x \in \mathcal{X}, y \in \mathcal{Y}} \left[ \nabla_{\bar{\theta}_L} y \log f_\theta(x) \cdot \left( \nabla_{\bar{\theta}_L} y \log f_\theta(x) \right)^T \right]$   `// ` $\bar{\theta}_L$ ` is the last layer's`
 `parameter of initial backdoor model.`
**for** $i = 1$ **to** $\mathcal{N}$ **do**
  Calculate loss, $\mathcal{L} = \mathcal{L}_{CE}(\mathcal{Y}, f_{\theta^i}(\mathcal{X})) + \frac{\eta}{2} \sum_j (\text{diag}(\bar{F}))_j \cdot (\theta^i_{L,j} - \bar{\theta}_{L,j})^2$
  $F \leftarrow \frac{1}{|\mathbb{D}_{val}|} \sum_{x \in \mathcal{X}, y \in \mathcal{Y}} \left[ \nabla_{\theta^i_L} y \log f_{\theta^i}(x) \cdot \left( \nabla_{\theta^i_L} y \log f_{\theta^i}(x) \right)^T \right]$       `// ` $\theta^i_L$ ` is the last`
   `layer's parameter at ` $i^{th}$ ` iterations`
  $\theta^{i+1}_L \leftarrow \theta^i_L - \alpha \cdot F^{-1} \nabla_{\theta^i_L}(\mathcal{L})$                      `// ` $\alpha$ ` is the learning rate`
  $\theta^{i+1} \leftarrow \{\mathbf{W}_{0,1}, \mathbf{W}_{1,2}, \cdots, \mathbf{W}_{L-2,L-1}, \theta^{i+1}_L\}$     `// ` $\mathbf{W}_{i,i+1}$ `'s are frozen parameters`
$\theta_p \leftarrow \{\mathbf{W}_{0,1}, \mathbf{W}_{1,2}, \cdots, \mathbf{W}_{L-2,L-1}, \theta^{\mathcal{N}}_L\}$   `// ` $\theta_p$ ` is the purified model's parameter`
**Output:** Purified Model, $f_{\theta_p}$

---

## C   MORE EXPLANATIONS ON SMOOTHNESS AND BACKDOOR

### C.1   WHY SMOOTHNESS ANALYSIS W.R.T CLEAN DISTRIBUTION?

In general, a backdoor model is usually well optimized w.r.t clean and poison data distribution. Therefore, it is designed to perform well on both distributions. If we look at the smoothness analysis of backdoor models *w.r.t.* original poisoned training data (both data distributions), the loss surface will be smoother. However, looking from only clean distribution point of view the loss surface is sharper as we have described in the paper. Backdoor purification implies that the model will only be sensitive to clean data distribution and completely ignore any type of backdoor manipulations. Since the model's behavior *w.r.t.* clean distribution is of our particular interest, we perform the smoothness

---

[2]https://github.com/kr-anonymous/ngf-animus

analysis *w.r.t.* clean distribution in all cases. If we put it another way, making the loss surface smooth *w.r.t.* clean distribution ensures that the model will automatically forget the poison distribution, *i.e.*, backdoor purification. Another minor reason is that, we are able to distinguish the behavior of benign and backdoor models because we consider clean distribution. This is the only common distribution between these models, and one has to perform smoothness analysis *w.r.t.* a common distribution to distinguish them.

## C.2 WHY SMOOTHNESS IS THE KEY TO REMOVING THE BACKDOOR?

One key observation from the smoothness study is that: there exists a key difference between weight-loss surface smoothness (estimated by *loss hessian*) of a backdoor and a benign model w.r.t. clean distribution—the weight-loss surface of a backdoor model is less smooth compared to a benign model. To further elaborate, let us consider feeding a clean sample to a backdoor model. By definition, it will predict the correct ground truth label. Now, consider feeding a sample with a backdoor trigger on it. The model will predict the adversary-set target label implying significant changes in prediction distribution. This significant change can be explained by the surface smoothness. In order to accommodate this significant change in prediction, the model must adjust itself accordingly. Such adjustment leads to non-smoothness in the weight-loss surface. *A non-smooth surface causes significant changes in loss gradient for specific inputs.* In our case, these specific inputs are backdoor-triggered samples. As the magnitude of a trigger is usually very small compared to the total input magnitude, the model has to experience quite a significant change in its weight space to cause large loss changes. We characterize this change in terms of smoothness. As for backdoor removal, we claim that making the non-smooth weight loss surface smoother removes the backdoor behavior. Based on the above discussion, a smoother surface should not cause a large change in loss or model predictions corresponding to backdoor related perturbations or triggers. In summary, for a model to show certain backdoor behavior, there are some specific changes that take place in the weight space. In this work, we try to explain these changes in terms of weight-loss surface smoothness. Our intuition is well supported by our comprehensive empirical evaluations.

## C.3 WHY USE FISHER INFORMATION MATRIX FOR ACHIEVING SMOOTHNESS?

In Fisher Information Matrix (FIM) based optimization, the natural gradient is defined as $F^{-1}\nabla L$ (ref. Eq. 5). From the perspective of information geometry, natural gradient defines the *direction in parameter space* which gives largest change in objective **per unit of change in model** ($p(y|x, \theta)$). Per unit of change in model is measured by KL-divergence. Note that KL-divergence is well connected with FIM as it can be used as a local quadrature approximation of KL-divergence of *model change*. Eqn. 2 suggests that one requires the knowledge of the original parameter ($\theta$) space to estimate it. Therefore, FIM can be thought of as a mechanism to translate between the geometry of the model ($p(y|x, \theta)$) and the current parameters ($\theta$) of the model. The way natural gradient defined the *direction in parameter space* is contrastive to the stochastic gradient. Stochastic gradient defines the direction in parameter space for largest change in objective **per unit of change in parameter** ($\theta$) measured by Eucludian distance. That is, the gradient direction is solely calculated based on the changes of parameters, without any knowledge of model geometry.

As FIM-based optimization minimizes the **changes in model**, the model itself cannot significantly change at each iteration. So, the overall optimization process goes through comparatively smoother transition and finally reaches smoother minima in comparison with SGD-based optimization[3].

## C.4 WHY DOES SAM UNDERPERFORMS?

One-layer optimization becomes a shallow network optimization problem for which there can exist many bad local minima. For such an optimization problem, typically first-order optimizers perform poorly mainly for the unawareness of loss surface curvature geometry. In the case of SAM, it uses SGD as the optimizer. Informally, the working principle of SAM is: at each iteration, SAM tries to minimize the maximum loss within a certain area in the loss weight space. Note that the formulation of finding the maximum loss in a certain area is based on 'Euclidean distance' metric which does not capture the curvature information of the plane. Although SAM performs better than vanilla SGD

---

[3]We refer the readers to (Jia & Su, 2020) for more discussion on FIM-based smoothness analysis.

in deep network[4] in terms of smoother optimization point, SAM's performance in shallow network (our case) is almost similar to vanilla SGD. Two potential reasons behind this poor performance are (i) using a predefined local area to search for maximum loss, and (ii) using 'Euclidean distance' metric instead of geometric distance metric. In contrast, NGD with curvature geometry aware Fisher Information Matrix can successfully avoid such bad minima and optimizes to a global minima.

## D  EXPERIMENTAL DETAILS

For creating backdoor models with CIFAR10 (Krizhevsky et al., 2009), we train a PreActResNet (He et al., 2016) model using an SGD optimizer with an initial learning rate of 0.01, learning rate decay of 0.1/100 epochs for 250 epochs. We also use a weight decay of $5e^{-4}$ with momentum of 0.9. We use a longer backdoor training to ensure a satisfactory attack success rate. We use a batch size of 128. For GTSRB (Stallkamp et al., 2011), we train a WideResNet-16-1 (Zagoruyko & Komodakis, 2016) model for 200 epochs with a learning rate of 0.01 and momentum of 0.9. We also regularize the weights with a weight-decay of $5e^{-4}$ We rescale each training image to $32 \times 32$ before feeding them to the model. The training batch size is 128 and an SGD optimizer is used for all training. We further created backdoor models trained on the Tiny-ImageNet and ImageNet datasets. For Tiny-ImageNet, we train the model for 150 epochs with a learning rate of 0.005, a decay rate of 0.1/60 epochs, and a weight decay of 1e-4. For ImageNet, we train the model for 200 epochs with a learning rate of 0.02 with a decay rate of 0.1/75 epochs. We also employ 0.9 and 1e-4 for momentum and weight decay, respectively. The details of these four datasets are presented in Table 7.

### D.1  DETAILS OF ATTACKS

We use 11 different attacks for CIFAR10. Each of them differs from each other in terms of either label mapping type or trigger properties. For label poisoning attack, we use a fixed poison rate of 10%. However, we need to increase this rate to 80% for CLB and SIG. For Blend and SIG attacks, we use a image-trigger mixup ratio of 0.2. WaNet adopts a universal wrapping augmentation as the backdoor trigger. Note that WaNet can be considered as an non-additive attack since it works like a augmentation technique with direct information insertion or addition like Badnets or TrojanNet. ISSBA adds specific trigger to each input that is of low magnitude and imperceptible. Both of these methods are capable of evading some existing defenses. For BPPA attack, we follow the PyTorch implementation[5]. For Feature attack, we create backdoor model based on this implementation [6]. Apart from clean-label attacks, we use a poison rate of 10% for creating backdoor attacks. The details of these attacks are presented in Table 8. In addition to theses attacks, we also consider 'All2All' attacks (Troj-all, Dyn-all) where we have more than one target label. To implement this attack, we change the given label $i$ to the target label $i + 1$. For class 9, the target label is 0.

Table 7: Detailed information of the datasets and DNN architectures used in our experiments.

| Dataset | Classes | Image Size | Training Samples | Test Samples | Architecture |
|---|---|---|---|---|---|
| CIFAR-10 | 10 | 32 x 32 | 50,000 | 10,000 | PreActResNet18 |
| GTSRB | 43 | 32 x 32 | 39,252 | 12,630 | WideResNet-16-1 |
| Tiny-ImageNet | 200 | 64 x 64 | 100,000 | 10,000 | ResNet34 |
| ImageNet | 1000 | 224 x 224 | 1.28M | 100,000 | ResNet50 |

### D.2  NGF AND OTHER OPTIMIZER IMPLEMENTATION DETAILS

To implement our proposed algorithm, we freeze the CNN backbone of the model and only fine-tune the linear or classification layer parameters. We perform the fine-tuning for 100 epochs with a learning rate of 0.01, weight decay of $1e^{-4}$, momentum of 0.9, and a batch size of 128. For studies

---

[4]Deep network consists of degenerate local minima and manifold of connect global minima (Liu et al., 2022) implying that, in deep network, there is no such bad local minima, unlike to shallow network, that could affect the performance of SAM.

[5]https://github.com/RU-System-Software-and-Security/BppAttack

[6]https://github.com/Megum1/DFST

Table 8: Details of different backdoor attacks we have defended against.

| Attacks | Trigger Type | Label Mapping | Description | Poison Rate | Target Label |
|---|---|---|---|---|---|
| Badnets (Gu et al., 2019) | Checker Board $3 \times 3$ | Label Poison | Triggers are placed at bottom left corner of images | 10% | 0 |
| CLB (Turner et al., 2018) | Checker Board $3 \times 3$ | Clean Label | use PGD-based adversarial perturbations | 80% | 0 |
| SIG (Barni et al., 2019) | Sinusoidal Signal | Clean Label | Use Mixup for adding the sinusoidal trigger to whole image | 80% | 0 |
| Dynamic (Nguyen & Tran, 2020) | Optimization | Label Poison | Generate image dependent triggers | 10% | 0 |
| Trojan (Liu et al., 2017) | Watermarks | Label Poison | Watermarks are static for all poisoned samples | 10% | 0 |
| Blend (Chen et al., 2017) | Random Pixels | Label Poison | Each pixel of the trigger is sampled from uniform distribution of [0,255] | 10% | 0 |
| CBA (Lin et al., 2020) | Mixer Constructor | Label Poison | Mixing existing benign features of two/more classes | 10% | 0 |
| FBA (Cheng et al., 2021) | Style Generator | Label Poison | Use a controlled detoxification to manipulate deep features | 10% | 0 |
| BPPA (Wang et al., 2022) | Quantization Trigger | Label Poison | Image quantization & contrastive adversarial learning based | 10% | 0 |

Table 9: Comparison of NGF with another state-of-the-art defense CLP (Zheng et al., 2022). Even though CLP achieves satisfactory removal performance for some attacks, the clean test accuracy drops significantly for some attacks (Blend, TrojanNet, CLB). We consider CIFAR10 dataset for this comparison.

| Methods | Badnets | | Blend | | Troj-one | | SIG | | CLB | | Dyn-One | | Dyn-All | | BPPA | |
|---|---|---|---|---|---|---|---|---|---|---|---|---|---|---|---|---|
| | ASR | ACC | ASR | ACC | ASR | ACC | ASR | ACC | ASR | ACC | ASR | ACC | ASR | ACC | ASR | ACC |
| No Defense | 100 | 92.96 | 100 | 94.11 | 100 | 89.57 | 100 | 88.64 | 100 | 92.78 | 100 | 92.52 | 100 | 92.61 | 99.70 | 93.82 |
| CLP | 2.58 | **90.90** | 0.81 | 81.17 | **1.81** | 56.08 | 1.46 | 53.03 | 13.61 | **90.38** | 13.84 | 90.84 | 14.78 | 89.72 | 11.39 | 90.21 |
| NGF | **1.86** | 88.32 | **0.38** | **91.17** | 2.64 | **84.21** | **0.12** | **84.16** | **1.04** | 88.37 | **1.17** | **90.97** | **1.61** | **90.19** | 7.14 | **91.84** |

with different optimizers (Adam, RMSProp, *etc.*), we use similar training settings as NGF. For sharpness-aware minimization, we restrict the search region for the SGD optimizer. We follow Pytorch implementation described here[7]. We use a batch size of 128 and a learning rate of 0.01 for SAM.

## D.3 DETAILS OF OTHER DEFENSES

For experimental results with ANP (Wu & Wang, 2021), we follow the source code implementation [8]. After creating each of the above mentioned attacks, we apply adversarial neural pruning on the backdoor model for 500 epochs with a learning rate of 0.02. We use the default settings for all attacks. For vanilla FT, we perform simple DNN fine-tuning with a learning rate of 0.01 for 125 epochs. We higher number of epochs for FT due to its poor clean test performance. The clean validation size is 1% for both of these methods. For I-BAU (Zeng et al., 2021), we follow their PyTorch Implementation [9] and purify the model for 10 epochs. We use 5% validation data for I-BAU. For AWM (Chai & Chen, 2022), we train the model for 100 epochs and use Adam optimizer with a learning rate of 0.01 and a wight decay of 0.001. We use the default hyper-parameter setting as described in their work $\alpha = 0.9, \beta = 0.1, \gamma = 108, \eta = 1000$. Above settings is for CIFAR10 and GTSRB only. For Tiny-ImageNet, we keep most of the training settings similar except reducing the number of epochs significantly. We also increase the validation size to 5% for vanilla FT, ANP, and AWM. For I-BAU, we use a higher valiadtion size of 10%. For purification, we apply ANP and AWM for 30 epochs, I-BAU for 5 epochs and Vanilla FT for 25 epochs. For ImageNet, we use a 3% validation size for all defenses (except for I-BAU, we use 5% validation data) and use different number of purification

---

[7] https://github.com/davda54/sam
[8] https://github.com/csdongxian/ANP_backdoor
[9] https://github.com/YiZeng623/I-BAU

Table 10: Correction rate (%) for different defense techniques. We define the correction rate (CR) as the percentage of poisonous samples correctly classified to their original classes. The higher the CR, the better is that method. We use CIFAR10 dataset for these evaluations.

| Method | Badnets | Trojan | CLB | SIG |
|---|---|---|---|---|
| No Defense | 0 | 0 | 0 | 0 |
| Vanilla FT | 85.74 | 80.52 | 84.72 | 43.35 |
| ANP | 85.56 | 80.69 | 82.04 | **45.64** |
| NGF (Ours) | **86.42** | **80.85** | **85.63** | 45.18 |

Table 11: Average run time for different defense methods. We consider CIFAR10 dataset and all attacks to calculate the average runtime. We do not show the runtime of CLP as it severely underperforms compared to other defenses. An NVIDIA RTX 3090 GPU was used for all computations.

| Method | Vanilla FT | ANP | I-BAU | AWM | NGF (Ours) | NGF w/o Regularizer (Ours) |
|---|---|---|---|---|---|---|
| Runtime (sec) | 78.1 | 201.5 | 52.7 | 90.2 | 38.3 | **37.8** |

epochs for different methods. We apply I-BAU for 2 epochs. On the other hand, we train the model for 3 epochs for ANP, AWM and vanilla FT.

## E    COMPARISON WITH ADDITIONAL DEFENSE

In Table 9, we show the comparison of NGF with a recently proposed defense technique based on channel Lipschitzness Pruning (CLP) (Zheng et al., 2022) that works without any data. We follow the *Github link*[10]. Based on the trigger-activated change on channel activation, CLP prunes channel. One disadvantage of pruning based method is that in case of challenging scenarios, *e.g.*,, strong attacks, large datasets and models *etc.*, it prunes neurons abruptly. This creates high possibility of pruning neurons sensible to clean data distribution. In turn, the clean test accuracy may decrease significantly in some scenario. As shown in Table 9, clean accuracies (ACCs) for Blend, Trojan and CLB attacks are much lower compared to NGF. Even though CLP performs reasonably well at removing backdoor, NGF still outperforms in that area.

## F    MORE ON SMOOTHNESS ANALYSIS

For smoothness analysis, we follow the PyHessian implementation[11] and modify it according to our needs. We use a single batch with size 200 to calculate the loss Hessian for all attacks with CIFAR10 and GTSRB datasets. We conduct further smoothness analysis for ImageNet dataset and different architectures. In Fig. 5, we show the Eigen density plots for different 5 different attacks. We used 2 A40 GPUs with 96GB system memory. However, it was not enough to calculate the loss hessian if we consider all 1000 classes of ImageNet. Due to GPU memory constraint, we consider ImageNet subset with 12 classes. We train a ResNet34 architecture with 5 different attacks. To calculate the loss hessian, we use a batch size of 50. Density plots before and after purification further confirms our proposed hypothesis. To test our hypothesis for larger architectures, we consider 5 different architectures for CIFAR10, *i.e.*, VGG19 (Simonyan & Zisserman, 2014), MobileNetV2 (Sandler et al., 2018), DenseNet121 (Huang et al., 2017), GoogleNet (Szegedy et al., 2014), Inception-V3 (Szegedy et al., 2016). Each of the architectures is deeper compared to the ResNet18 architecture we consider for CIFAR10. Due to their large size, showing the effectiveness of NGF in case of these architecture will strengthen our claim—*one layer NGF based fine-tuning is enough for backdoor purification.* In Fig. 6, we show the performance of NGF when backdoor models are created using these architectures. Our proposed one layer fine-tuning successfully removes the backdoor in all of these scenarios.

---

[10] https://github.com/rkteddy/channel-Lipschitzness-based-pruning
[11] https://github.com/amirgholami/PyHessian

Table 12: Performance of NGF while fine-tuning all layers of DNN. The results shown here are for CIFAR10 dataset.

| Methods | Badnets ASR | Badnets ACC | Blend ASR | Blend ACC | Trojan ASR | Trojan ACC | Dynamic ASR | Dynamic ACC | CLB ASR | CLB ACC | SIG ASR | SIG ACC | CBA ASR | CBA ACC | Runtime (Secs.) |
|---|---|---|---|---|---|---|---|---|---|---|---|---|---|---|---|
| Initial | 100 | 92.96 | 100 | 94.11 | 100 | 89.57 | 100 | 92.52 | 100 | 92.78 | 100 | 88.64 | 93.20 | 90.17 | – |
| Vanilla-FT (All Layers) | 4.87 | 85.92 | 4.77 | 87.61 | 3.78 | 82.18 | 4.73 | 88.61 | 1.83 | 87.41 | 1.04 | 81.92 | 27.80 | 83.79 | 78.1 |
| NGF (Last layer) | 1.86 | 88.32 | **0.38** | 91.17 | 2.64 | 84.21 | 1.17 | **90.97** | 1.04 | 88.37 | **0.12** | 84.16 | 24.60 | 85.97 | **38.3** |
| NGF (All layers) | **1.47** | **88.65** | 0.42 | **92.28** | **2.05** | **84.61** | **1.06** | 90.42 | **0.60** | **88.74** | 0.18 | **85.12** | **19.86** | **86.30** | 173.2 |

Table 13: Purification performance after randomly re-initializing the last layer. Even after re-initialization, the purification task is similar as before, *i.e.*, *proper fine-tuning*. Without proper fine-tuning, the backdoor behavior will be still present after purification. In contrast to SGD, NGF is highly successful even after such re-initialization. CIFAR10 dataset is considered here.

| | BadNets ASR | BadNets ACC | Trojan ASR | Trojan ACC | Blend ASR | Blend ACC | CLB ASR | CLB ACC | SIG ASR | SIG ACC |
|---|---|---|---|---|---|---|---|---|---|---|
| No Defense | 100 | 92.96 | 100 | 89.57 | 100 | 94.11 | 100 | 92.78 | 100 | 88.64 |
| SGD | 93.51 | 89.35 | 89.63 | **87.55** | 71.22 | **92.21** | 88.01 | **90.01** | 76.83 | **86.12** |
| NGF | **3.34** | **89.65** | **18.78** | 84.21 | **0.33** | 86.39 | **4.45** | 82.50 | **1.25** | 83.95 |

## G LABEL CORRECTION RATE OF DIFFERENT DEFENSES

In standard removal measurement, it is sufficient for backdoored images to be classified as a non-target class. As we calculate ASR after removal, our evaluation follows standard measurement. We define the correction rate (CR) as the percentage of poisonous samples correctly classified to their original classes. We define the method with the highest value of CR as the best performing or SOTA method. We use the CIFAR10 dataset and 4 different attacks for demonstration. It can be observed from Table 10 that our method obtains SOTA correction performance for most of these attacks.

## H RUNTIME ANALYSIS OF OTHER DEFENSES

In Table 11, we show the average runtime of other defenses. It can be observed that ANP is almost 6x slower than NGF. Other defenses, NAD and MCR are also much slower than NGF. NAD uses transfer learning based distillation using a teacher-student framework. However, the complexity of this method results in computational overhead. Instead, NGF revisits much simpler fine-tuning approach from one-layer optimization point of view. Our simple and effective method leads to one of the fastest purification. We take the average of all run times against 11 attacks on CIFAR10. Note that, for each epoch in NGF, we have to feed-forward all validation data. However, we only update the parameters of last layer through back-propagation. The reason behind this is that we use different data augmentations while fine-tuning. This does not allow us to save the CNN features one time and re-use it for upcoming all epochs.

## I FINE-TUNING ALL LAYERS

We have considered fine-tuning all layers fusing NGF and SGD. Note that vanilla FT does fine-tune all layers. We report the performance of NGF for all layers in Table 12. While fine-tuning all layers seems to improve the performance, it takes almost $6\times$ more computational time than NGF on last layer. We perform NGF with the regularizer here.

Table 14: Purification performance for various validation data size. NGF performs well even with very few validation data, *e.g.*, 50 data points. All results are for CIFAR10 and Badnets attack.

| Validation size | 50 | | 100 | | 250 | | 350 | | 500 | |
|---|---|---|---|---|---|---|---|---|---|---|
| Method | ASR | CA | ASR | CA | ASR | CA | ASR | CA | ASR | CA |
| No Defense | 100 | 92.96 | 100 | 92.96 | 100 | 92.96 | 100 | 92.96 | 100 | 92.96 |
| ANP | 13.66 | 83.99 | 8.35 | 84.47 | 5.72 | 84.70 | 3.78 | 85.26 | 2.84 | 85.96 |
| AWM | 8.51 | 83.63 | 7.38 | 83.71 | 5.16 | 84.52 | 5.14 | 85.80 | 4.34 | 86.17 |
| NGF (Ours) | **6.91** | **86.82** | **4.74** | **86.90** | **4.61** | **87.08** | **2.45** | **87.74** | **1.86** | **88.32** |

Table 15: Illustration of purification performance for All2All attack using CIFAR10 dataset, where uniformly distribute the target labels to all available classes. NGF shows better robustness and achieves higher clean accuracies for 3 attacks: Badnets, Blend, BPPA with 10% poison-rate.

| Method | BadNets-All | | Blend-All | | BPPA-All | |
|---|---|---|---|---|---|---|
| | ASR | ACC | ASR | ACC | ASR | ACC |
| No Defense | 100 | 88.34 | 100 | 88.67 | 99.60 | 92.51 |
| FT | 2.78 | 83.19 | 2.83 | 80.13 | 10.97 | 89.76 |
| NAD | 4.58 | 81.34 | 6.76 | 81.13 | 20.19 | 87.77 |
| ANP | 3.13 | 82.19 | 4.56 | 82.88 | 9.87 | 89.91 |
| NGF (Ours) | **1.93** | **84.29** | **1.44** | **83.79** | **6.10** | **90.56** |

Table 16: Performance of NGF against combined backdoor attack. We poison some portion of the training data using 3 different attacks; Badnets, Blend, and Trojan. Each of these attacks have an equal share in the poison data. All results are for CIFAR10 datasets containing different number of poisonous samples.

| Poison Rate | 10% | | 25% | | 35% | | 50% | |
|---|---|---|---|---|---|---|---|---|
| Method | ASR | ACC | ASR | ACC | ASR | ACC | ASR | ACC |
| No Defense | 100 | 88.26 | 100 | 87.51 | 100 | 86.77 | 100 | 85.82 |
| MCR | 27.83 | 78.10 | 31.09 | 77.42 | 36.21 | 75.63 | 40.08 | 72.91 |
| ANP | 4.75 | 83.50 | 5.42 | **81.73** | 6.51 | 79.93 | 9.76 | 78.06 |
| NGF (Ours) | **1.17** | **83.61** | **2.15** | 81.62 | **3.31** | **80.01** | **4.15** | **79.35** |

## J    EFFECT OF CLEAN VALIDATION DATA SIZE

We also present how the total number of clean validation data can impact the purification performance. In Table 14, we see the change in performance while gradually reducing the validation size from 1% to 0.1%. We consider Badnets attack on CIFAR10 dataset for this evaluation. Even with only 50 (0.1%) data points, NGF can successfully remove the backdoor by bringing down the attack success rate (ASR) to 6.91%. We also consider adversarial weight masking for this comparison. For both ANP and AWM, reducing the validation size has severe impact on the test accuracy (ACC).

## K    PURIFICATION USING LAST LAYER RE-INITIALIZATION

We also conduct studies on the behavioral difference of SGD and NGD while we re-initialize the last layer. Even though we re-initialize the last layer, one still has to properly fine-tune the backdoor model to remove the backdoor. However, we see throughout our evaluations that SGD-based one-layer fine-tuning is not a proper fine-tuning method and unable to remove backdoor. In case of re-initialization, the shallow network optimization problem still stands as wells as the issue of bad local minima. Therefore, SGD shows similar behavior in this scenario too. Table 13 shows the performance of SGD and NGF in case one decides to re-initialize the layer than fine-tunes. As usual, NGF is able to reach smooth minima due to its ability to properly fine-tune the model.

## L    MORE ALL2ALL ATTACKS

Most of the defenses evaluate their methods on only All2One attack where we consider only one target label. However, there can be multiple target classes in a practical attack scenario. We consider one such case: All2All attack where target classes are uniformly distributed among all available classes. In Table 15, we show the performance under such settings for 3 different attacks with a poison rate of 10%. It shows that All2All attack is more challenging to defend against as compared to All2One attack. However, the performance of NGF seems to be consistently better than other defenses for both of these attack variations. For reference, we achieve an ASR improvement of 3.12% over ANP while maintaining a lead in classification accuracy too.

## M    COMBINING DIFFERENT BACKDOOR ATTACKS

We also perform experiments with combined backdoor attacks. To create such attacks, we poison some portion of the training data using 3 different attacks; Badnets, Blend, and Trojan. Each of these

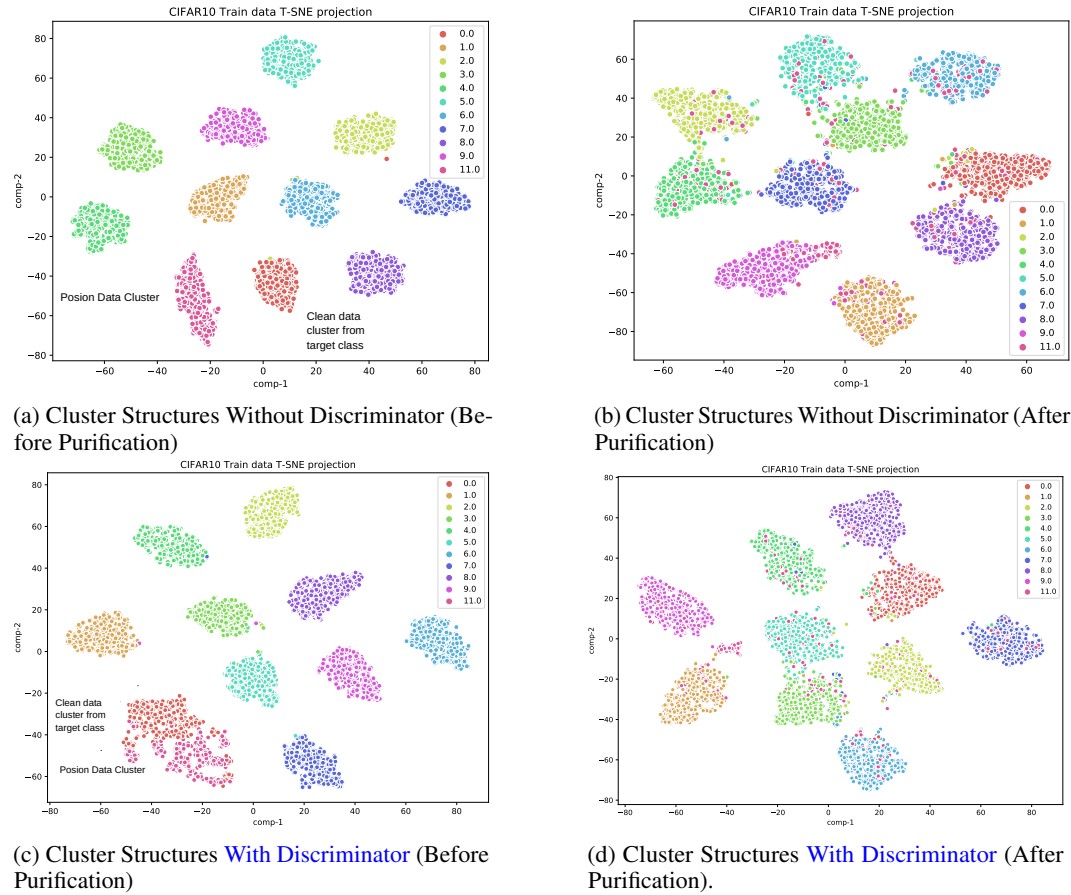

(a) Cluster Structures Without Discriminator (Before Purification)

(b) Cluster Structures Without Discriminator (After Purification)

(c) Cluster Structures With Discriminator (Before Purification)

(d) Cluster Structures With Discriminator (After Purification).

Figure 3: t-SNE visualization of class features for CIFAR10 dataset with Badnets attack. For visualization purpose only, we assign label "0" to clean data cluster from target class and label "11" to poison data cluster. However, both of these clusters have same training label "0" during training. With discriminator 3c, the clusters are relatively closer and harder to discriminate as compared to without discriminator 3a. However, after fine-tuning using clean data, NGF can remove the backdoor effect. Even though the clusters are very close, the classifier can still discriminate the clusters as shown by the 3b&3d. After purification, poison data are distributed among their original ground truth classes instead of the target class. To estimate these clusters after purification (3b&3d), we take the output of classifier before softmax (embedding dim=10) and apply tSNE with 2 components.

attacks have an equal share in the poison data. As shown in Table 16, we use 4 different poison rates: $10\% \sim 50\%$. NGF outperforms other baseline methods (MCR and ANP) by a satisfactory margin.

## N    DECISION HEATMAPS: HOW NGF REMOVES BACKDOOR?

While inserting the backdoor behavior, the model, especially the linear classification layer, memorizes the poison data distribution. By memorization, we mean it memorizes the simpler trigger pattern. Whenever the model sees that pattern in the input, it prioritizes the trigger-specific feature instead of image-specific (clean part) feature and predicts the adversary-set target label. When we re-train or fine-tune the classifier with clean validation data, the classifier forgets the poison distribution as fine-tuning reinforces the dominance of clean features in model prediction. After fine-tuning, the model looks for image-specific features for prediction as it has almost no memory of the trigger-specific features. We illustrate the decision heat-maps for clean, backdoor and purified model in Figure 4. We show the decision heatmaps for clean an poison data. As clean model is only trained on clean data, it is not sensitive to the trigger. Our defense objective says that, a purified model should behave like a benign model, *i.e.*, the decision making process (for clean and poison data) should resemble a clean

model. As we can see for the poison data, NGF successfully removes the effect of the trigger. The purified model ignores the trigger while making decisions.

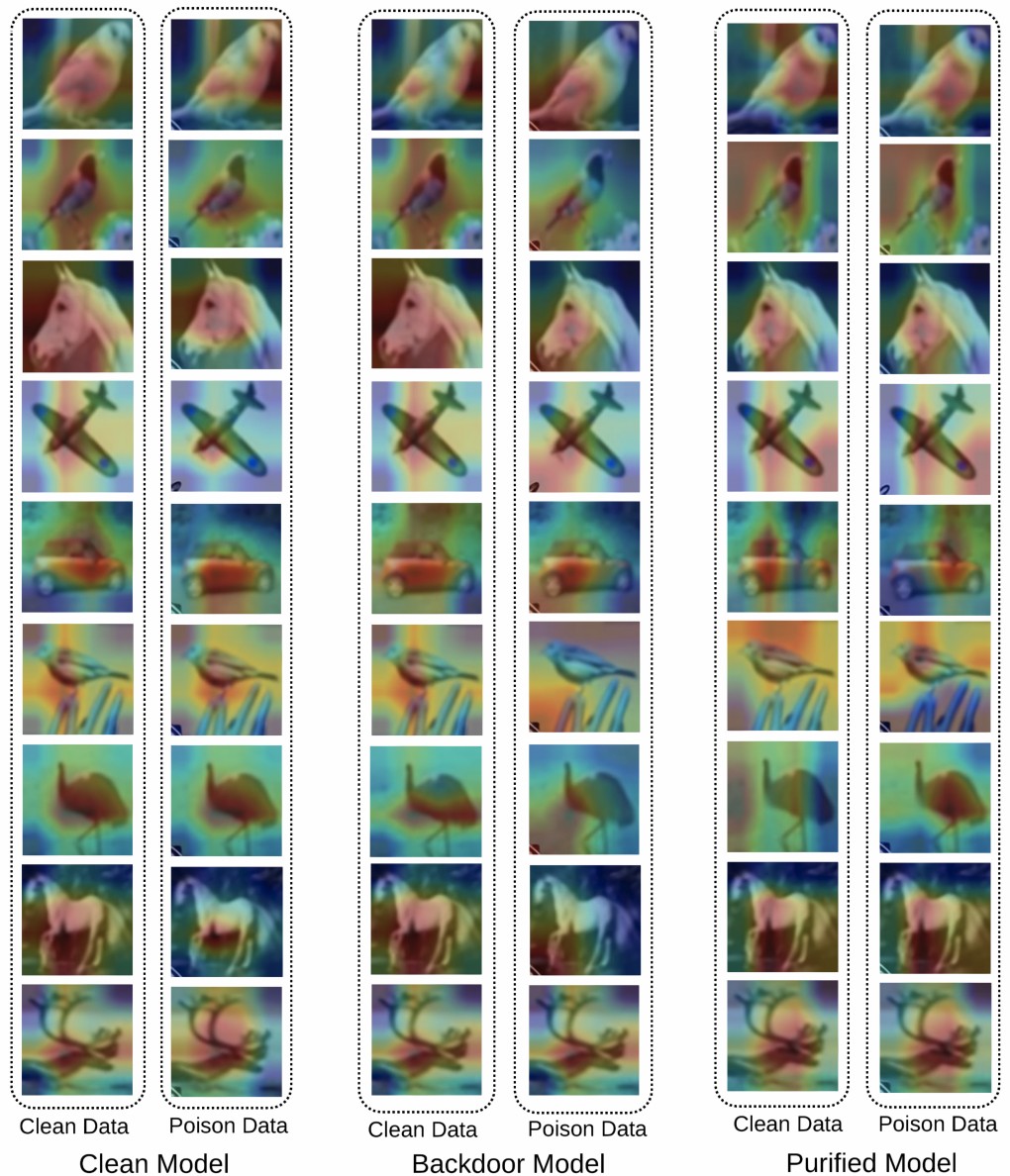

Clean Data   Poison Data      Clean Data   Poison Data      Clean Data   Poison Data

**Clean Model**         **Backdoor Model**         **Purified Model**

Figure 4: Decision heat-maps for clean, backdoor and purified models. Regions with more reddish color is more responsible towards a decision making. For each category, we show the heatmaps for clean and poison data. *Trigger is at bottom left corner of each poison data*. Unlike backdoor model, clean model is insensitive to triggers in the poison sample. Wheres backdoor model causes the model to make wrong decision based on the trigger pattern. The purified model behaves like a clean model and does not look at the trigger while making a decision. All heat-maps are generated for CIFAR10 dataset attacked with BadNets. We choose this attack for better understanding of the context.

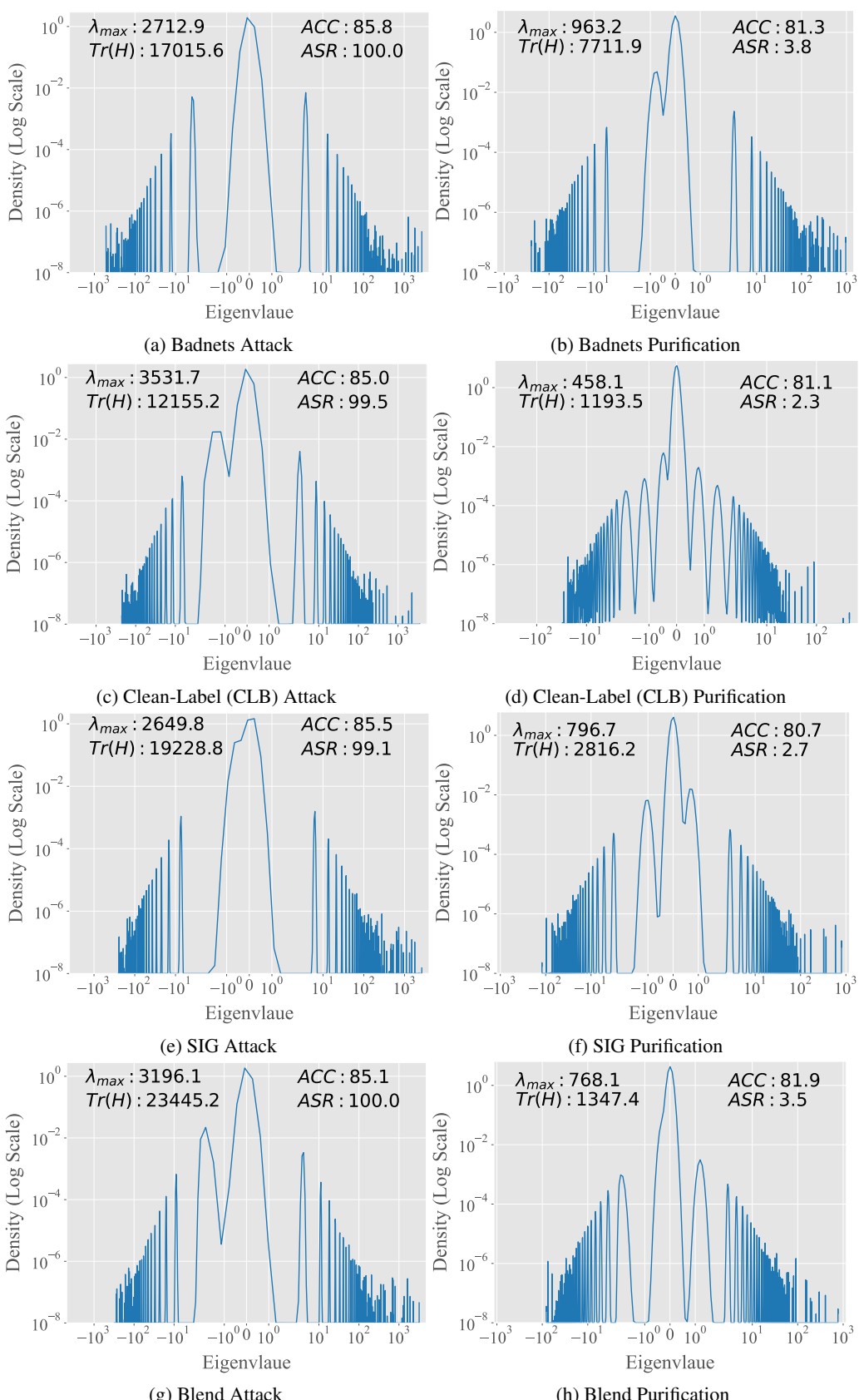

Figure 5: Smoothness analysis for ImageNet Subset (first 12 classes). A ResNet34 architecture is trained on the subset. For GPU memory constraint, we consider only first 12 classes while calculating the loss Hessian. Eigen Density plots of backdoor models (before and after purification) are shown here.

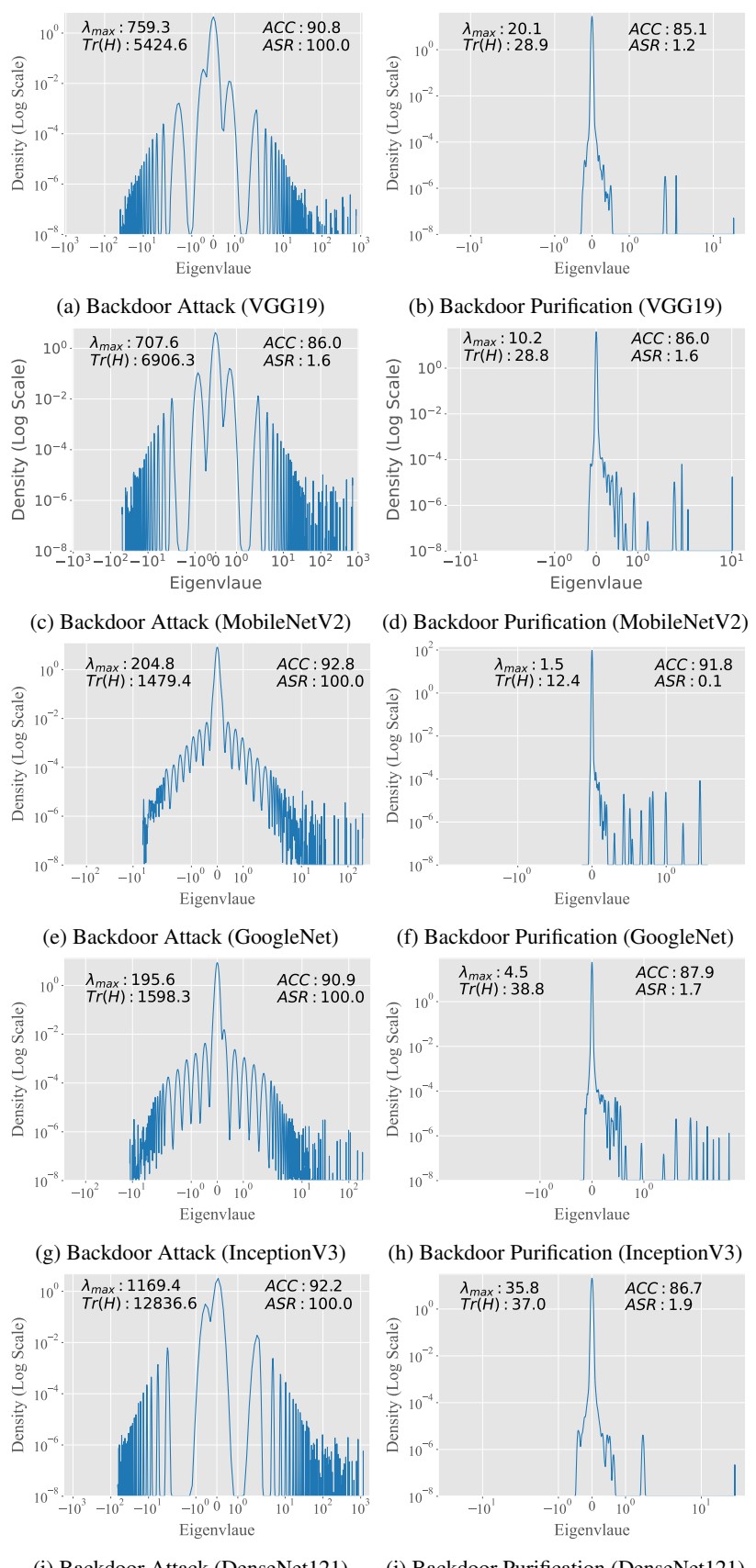

Figure 6: Smoothness Analysis for different architectures. For all architectures, we consider badnets attack on CIFAR10.

