# OpenReview forum: "In Search of Smooth Minima for Purifying Backdoor in Deep Neural Networks"
_ICLR.cc/2023/Conference — Submitted to ICLR 2023_

### Official Review · Reviewer_1UnP · 2022-10-24

**Confidence:** 3
**Correctness:** 3
**Technical Novelty And Significance:** 3
**Empirical Novelty And Significance:** 3
**Recommendation:** 6

**Clarity, Quality, Novelty And Reproducibility:**

Why does the natural choice, SAM, fails in defense, during the natural gradient descent success? This makes me still concerned about the correctness of the motivation (See weakness 4). Is the success of the proposed method genuinely related to the sharp loss landscape caused by poisoned data?

**Strength And Weaknesses:**

**Strength**
1. The paper demonstrates a side effect of backdoor attacks. When calculated only on benign data, the loss landscape of the minimum point becomes sharper. This might inspire further studies.
2. The proposed method focuses on fine-tuning of last linear layer, which influences fewer parameters and consumes less computation. The empirical results also show notable improvements compared to previous SOTA methods, which verifies the effectiveness of the proposed method.
3. Overall, the paper is easy to follow.

**Weakness**
1. What does $p$ in $\mathcal{L}_p$ mean? Considering the author only provides an approximation of $\mathcal{L}_p$,
what is the exact formulation?

2. In my opinion,  fine-tuning only to the linear layer has very limited ability, which makes me worried the performance. Although the authors show that the triggered data (images with trigger pattern) are not classified as the target class after defense (i.e., low ASR),  I am curious whether these triggered test data are classified into their ground-truth label.

3. I think the success of the proposed method relies on the separation between (1) the cluster of benign data from the target class and (2) the cluster of poisoned data. Since they are separated, we can use a linear classifier to distinguish them.
However, it might be easy to propose an adaptive attack, i.e., If the adversary is aware of the presence of the proposed defense, he can conduct an adaptive attack during backdoor attacks to escape. Specifically, he can introduce an extra linear discriminator on the top of the backbone in parallel with the classifier. The discriminator tries to predict whether the sample from the target class is benign or poisoned, while the classifier still predicts which class the sample belongs to. During training, he maximizes the discrimination loss while minimizing the classification loss.
After training, since the discriminator cannot predict whether the sample is benign or poisoned anymore, the cluster of benign data from the target class is heavily mixed with the cluster of poisoned data. At this time, does the proposed method still works?

4. Why does SAM fails in defense? The proposed method is inspired by the sharpness of the landscape caused by the additive poisoned data during training, while SAM is the most natural choice to recover the flatness. If SAM fails, does the success of the proposed method come from something else instead of the flatness?

5. The natural gradient descent (NGD) contributes the most improvements, and the proposed regularization only introduces little improvements in natural accuracy (see Table 5). Considering NGD has been widely applied, the technical novelty in this paper is limited.

Minor:
- The $\theta$ should be $\theta = [ W_{0, 1}, W_{1,2}, W_{L-1, L} ]$ above Equation (3), i.e., starting from $0$. Otherwise, there are only L-1 layers in total.

- In Equation (1) and Equation (3), the comma should follow the formula in the same line, rather than appearing in the front of the following sentence.


**Summary Of The Paper:**

This paper studies the parametric landscape in terms of the loss on clean data. Inspired by the phenomenon in the landscape, the authors propose to use natural gradient fine-tuning only to the last classifier layer for defense and also introduce clean data distribution-aware regularizer to keep the natural accuracy. Empirical results demonstrate the effectiveness of the proposed method.


**Summary Of The Review:**

This paper introduces an interesting side effect caused by poisoned data and proposes a simple but promising method to defend against backdoors attacks. However, there are still some concerns about adaptive attacks and the correctness of the motivation. If the authors could address my concern, I will reconsider my score.

---

> ### Author Response · Authors · 2022-11-08
> **Response to Reviewer 1UnP (2/2)**
>
> **Revierwer’s Comment:** *Why does SAM fails in defense? The proposed method is inspired by the sharpness of the landscape caused by the additive poisoned data during training, while SAM is the most natural choice to recover the flatness. If SAM fails, does the success of the proposed method come from something else instead of the flatness?*
>
> **Our Response:**
>
> One-layer optimization becomes a shallow network optimization problem for which there can exist many bad local minima. For such an optimization problem, typically first-order optimizers perform poorly mainly for the unawareness of loss surface curvature geometry. In the case of SAM, it uses SGD as the optimizer. Informally, the working principle of SAM is: at each iteration, SAM tries to minimize the maximum loss within a certain area in the loss weight space. Note that the formulation of finding the maximum loss in a certain area is based on ‘Euclidean distance’ metric which does not capture the curvature information of the plane. Although SAM performs better than vanilla SGD in deep network$^1$ in terms of smoother optimization point, SAM’s performance in shallow network (our case) is almost similar to vanilla SGD. Two potential reasons behind this poor performance are (i) using a predefined local area to search for maximum loss, and (ii) using ‘Euclidean distance’ metric instead of geometric distance metric. In contrast, NGD with curvature geometry aware Fisher Information Matrix can successfully avoid such bad minima and optimizes to a global minima.
>
> $^1$ Deep network consists of degenerate local minima and manifold of connect global minima [1] implying that, in deep network, there is no such bad local minima, unlike to shallow network, that could affect the performance of SAM.
>
> [1] Liu et al. "Loss landscapes and optimization in over parameterized non-linear systems and neural networks." *Applied and Computational Harmonic Analysis, 2022.*
> ****
>
> **Reviewer’s Comment:** *The natural gradient descent (NGD) contributes the most improvements, and the proposed regularization only introduces little improvements in natural accuracy (see Table 5). Considering NGD has been widely applied, the technical novelty in this paper is limited.*
>
> **Our Response:**
>
> As shown in Table 5 of our main paper, the clean-distribution aware regularizer boosts the ACC by 1% which is significant considering NGF achieves a total gain of 0.62% in ACC compared to I-BAU, *please refer to the CIFAR10 dataset in Table 2 of our main paper*. Without the regularizer, NGF might not have achieved state-of-the-art ACC. Therefore, the regularizer plays a very important role in establishing the superiority of our overall method to other baselines.
>
>
> ****
> ****
> *Additional note on how NGF-based fine-tuning works:* While inserting the backdoor behavior, the model, especially the linear classification layer, memorizes the poison data distribution. By memorization, we mean it memorizes the simpler trigger pattern. Whenever the model sees that pattern in the input, it prioritizes the trigger-specific feature instead of image-specific (clean part) feature and predicts the adversary-set target label. When we re-train or fine-tune the classifier with clean validation data, the classifier forgets the poison distribution as fine-tuning reinforces the dominance of clean features in model prediction. After fine-tuning, the model looks for image-specific features for prediction as it has almost no memory of the trigger-specific features. We have added a decision heatmap for predictions, before and after purification, in Appendix N (Figure 4).
>
> ***
> We thank the reviewer again and hopefully, our discussions make the reviewer’s concerns clear. We would be happy to address any further concerns.

---

> > ### Author Response · Authors · 2022-12-01
> > **Follow Up. Thank You for Your Review Again!**
> >
> > Dear Reviewer,
> >
> > We genuinely appreciate your positive feedback once again! Your constructive feedback properly guided us to revise the paper and made it much better. This is a friendly reminder that the final stage of the discussion period is drawing close. We would really appreciate it if you could let us know whether your concerns have been addressed and if we can clarify anything else.
> >
> > Nevertheless, Thank you for your kind efforts :),
> >
> > Authors of Paper 2286

---

> > > ### Comment · Reviewer_1UnP · 2022-12-06
> > > **Thanks for the reply**
> > >
> > > The replies are detailed and informative. After reading, I still have some problems.
> > >
> > > 1. Why do we only apply SAM to the last layer? Considering the sharpness is calculated with respect to all weight parameters in Sec 4, i.e., $\nabla_{\theta}^2 \mathcal{L}$, it is more natural and fair to apply SAM to all weight parameters rather than only parameters of the last layer.
> > > 2. Is the visualized feature the inputs to the last layer (representatives before the last layer) or the outputs of the last layer (representatives after the last layer) in Figure 3?

---

> > > > ### Author Response · Authors · 2022-12-06
> > > > **Thank You! Response to Further Comments**
> > > >
> > > > We thank the reviewer for further comments. Our responses are given below:
> > > >
> > > > *****
> > > >
> > > > **Response to Comment 1:**
> > > >
> > > > For all optimization techniques in Table 3 (NGF, SGD, SAM, Adam, etc.), we have measured smoothness w.r.t. full network parameters even though we only change the last layer. Therefore, the evaluation should be fair in this regard.
> > > >
> > > > Note that vanilla FT finetunes all parameters using SGD optimizer and *applying SAM to all parameters is essentially the same as “vanilla FT with SAM”*. For evaluating vanilla FT with SAM, we have considered 6 different attacks on CIFAR10 dataset and present the results in Table A here.
> > > >
> > > > It can be observed that our proposed method obtains better performance in terms of both attack success rate (ASR) and clean test accuracy (ACC). Similar to SGD-based vanilla FT, employing SAM for all parameters hurts the ACC significantly as changing all parameters often does that. Table A also shows that SAM has a slightly better ASR performance compared to SGD, which aligns with our smoothness hypothesis as SAM usually leads to smoother loss surface. Note that, we have discussed in detail (in the paper as well as our previous responses) why SAM does not obtain the same type of smoothness for one-layer fine-tuning.  As for execution time, each SAM update requires 2 backpropagation operations while non-SAM update (SGD, Adam, etc.) requires only 1 backpropagation. This makes vanilla FT (SAM) slower than vanilla FT (SGD) which is not desirable for backdoor purification techniques.
> > > >
> > > > **Table A: Comparison of NGF with Vanilla FT using SAM optimizer. We fine-tune all parameters for Vanilla FT (SAM) while finetuning only the last layer parameters for the proposed NGF. We consider CIFAR10 dataset for all evaluations. All entries are formatted as ASR/ACC.**
> > > > |Method|Badnets  |  Blend     | Troj-one   |   CLB      |   WaNet  |  ISSBA|
> > > > |:----------|:----------|:----------|:----------|:----------|----------:|----------:|
> > > > |No Defense  |100/92.96 | 100/94.11 | 100/89.57 | 100/92.78| 98.64/92.29|99.80/92.80|
> > > > |Vanilla FT (SGD)  |4.87/85.92| 4.77/87.61| 3.78/82.18| 1.83/87.41| 5.81/86.70| 6.76/85.42|
> > > > |Vanilla FT (SAM)  |3.91/85.75| 2.74/88.26| 3.53/82.52| 1.47/86.30| 4.24/85.90| 6.14/86.27|
> > > > |NGF (Ours) |**1.86/88.32**| **0.38/91.17**| **2.64/84.21** | **1.04/88.37**|  **2.38/89.58**|**4.24/90.18**|
> > > >
> > > > ******
> > > >  **Response to Comment 2:**
> > > >
> > > > For visualizing clusters *before purification (in Fig. 3(a) and 3(c))*, we take the inputs to the last layer (representatives before the last layer). We do not do the same for *after purification (in Fig. 3(b) and 3(d))* as we only finetune the last layer while keeping the CNN backbone unchanged. Therefore, after purification, we take the outputs of the last layer (representatives after the last layer).
> > > >
> > > > If we take the inputs to the last layer (representatives before the last layer) for visualizing clusters after purification, the cluster structures are the same as before purification (Fig. 3(a) and 3(c)) which is expected.
> > > >
> > > > *****
> > > > Thanks again for your time and kind effort in reviewing our paper. Please let us know if you have further concerns.

---

> > > > > ### Comment · Reviewer_1UnP · 2022-12-08
> > > > > **Thank you!**
> > > > >
> > > > > 1. It solves my concern about whether SAM works. Since the loss landscape inspires this paper and SAM is a direct solution to improve the sharpness, it is natural to evaluate SAM. The success of SAM on all parameters verifies the correctness of the motivation. I still recommend authors include the results of SAM on all parameters in Table 3. It will be more consistent with readers' intuition (i.e., SAM should work in this case, at least for all parameters).
> > > > >
> > > > > 2. Since we always visualize the feature vectors of the same layer, the difference between Figure 3(a) and Figure 3(b) confuses me. Your answer addresses my question.

---

> > > > > > ### Author Response · Authors · 2022-12-08
> > > > > > **Thank You Again for Your Prompt Feedback!**
> > > > > >
> > > > > >
> > > > > > Thank you for your reply! We are happy that we could address your concerns. We will definitely add the results of SAM on all parameters in Table 3. Since we cannot add this right away (as the deadline for uploading the revised version has already passed!), we will do it in the near future as soon as we are given the chance. We greatly appreciate your valuable time and efforts in making our submission much better! *A kind request to the reviewer:* since your concerns have been addressed, could you please reconsider your score? :)
> > > > > >
> > > > > >
> > > > > > Thanks,
> > > > > >
> > > > > > Authors

---

> > > > > > > ### Comment · Reviewer_1UnP · 2022-12-08
> > > > > > > **Thank you for the detailed replies**
> > > > > > >
> > > > > > > Since the authors have addressed my concerns, I have changed my rating from 5 to 6.

---

> > > > > > > > ### Author Response · Authors · 2022-12-08
> > > > > > > > **Thank You!**
> > > > > > > >
> > > > > > > > Dear Reviewer,
> > > > > > > >
> > > > > > > > Thank you so much for your kind consideration in raising the score.
> > > > > > > >
> > > > > > > > Best Regards,
> > > > > > > >
> > > > > > > > Authors

---

> ### Author Response · Authors · 2022-11-08
> **Response to Reviewer 1UnP (1/2)**
>
> Thank you for your valuable feedback. We addressed the reviewer’s concerns as follows:
>
> **Reviewer’s Comment:** *What does p in Lp  mean? Considering the author only provides an approximation of Lp, what is the exact formulation?*
>
> **Our Response:**
>
> $L_p$ is the loss for purification objective where $p$ stands for purification. We apologize for the ambiguity of the notation.
> Ideally, (cross-entropy) loss $\mathcal{L}(y,f_\theta(x))$ is calculated for the input (i.e., validation data) distribution. However, as we approximate the loss using input samples, which is a standard practice, we use the approximation symbol. In practice, computing exact loss for input distribution is intractable.
>
> ***
>
> **Reviewer’s Comment:** *In my opinion, fine-tuning only to the linear layer has very limited ability, which makes me worried the performance. Although the authors show that the triggered data (images with trigger pattern) are not classified as the target class after defense (i.e., low ASR), I am curious whether these triggered test data are classified into their ground-truth label.*
>
> **Our Response:**
>
> We thank the reviewer for this great point. We addressed this valid concern in our original submission , Appendix G (Table 10). In addition to ASR, we use another metric *Label Correction Rate (CR)*. *CR* is defined as the percentage of poisonous samples correctly classified to their original ground truths. We define the method with the highest value of CR as the best performing or SOTA method. We use CIFAR10 dataset and 4 different attacks for this evaluation. It can be observed from Table 10 that our method obtains SOTA label correction performance for most of these attacks.
>
> ****
>
> **Reviewer’s Comment:** *I think the success of the proposed method relies on the separation between (1) the cluster of benign data from the target class and (2) the cluster of poisoned data. Since they are separated, we can use a linear classifier to distinguish them. However, it might be easy to propose an adaptive attack, i.e., If the adversary is aware of the presence of the proposed defense, he can conduct an adaptive attack during backdoor attacks to escape. Specifically, he can introduce an extra linear discriminator on the top of the backbone in parallel with the classifier. The discriminator tries to predict whether the sample from the target class is benign or poisoned, while the classifier still predicts which class the sample belongs to. During training, he maximizes the discrimination loss while minimizing the classification loss. After training, since the discriminator cannot predict whether the sample is benign or poisoned anymore, the cluster of benign data from the target class is heavily mixed with the cluster of poisoned data. At this time, does the proposed method still works?*
>
>
> **Our Response:**
>
> We appreciate the reviewer’s feedback on the future aspects of our defense. The suggested approach (of the reviewer) can lead to an interesting future attack technique that may evade the proposed defense, NGF. Following reviewer’s suggestions, we have implemented the adaptive attack. For discriminator, we have used another parallel linear layer with Sigmoid activation. We use binary cross-entropy loss for the discriminator and regular cross-entropy loss for the original classification layer. The discriminator loss tries to differentiate between the clean data cluster from target class and poison data cluster. Whereas the classifier tries to classify the samples to their given labels. If we have a high attack success rate, that means the classifier is successful and the discriminator cannot differentiate anymore. We consider Badnets, Blend, and Trojan attacks on CIFAR10 for this evaluation. A poison rate of 10% has been employed for each of these attacks. After training for 100 epochs, we have ASR close to 100% as shown in Table A. As shown, NGF can successfully remove the backdoor even against a carefully designed adaptive attack.
>
> **Table A: Performance of NGF against adaptive attack suggested by the reviewer. All entries in the table are in ASR/ACC format.**
> |Defense Method|Badnets|Blend |Trojan |
> |:-|:-|:-|:-:|
> |No Defense|100/90.58|100/92.20|100/87.10|
> |NGF|2.56/87.16|0.94/90.45|1.68/83.72|
>
> We also show the t-SNE visualization of the model before and after purification. Please see Figure 3 of the updated appendix.

---

### Official Review · Reviewer_6Zjq · 2022-10-24

**Confidence:** 3
**Clarity, Quality, Novelty And Reproducibility:** The quality, clarity and originality …
**Correctness:** 3
**Technical Novelty And Significance:** 3
**Empirical Novelty And Significance:** 3
**Recommendation:** 6

**Strength And Weaknesses:**

Pros:
+ It is interesting to relate the model attack with optimization. The basic assumption of trapping in local minimum is convincing.

+ The proposed NGF is inspired by the Fisher Information Matrix, which has solid theoretical foundations to support it.

+ The entire manuscript is well written, with clear logic to follow.

+ There are solid and comprehensive experimental analyses throughout the paper. The proposed method has achieved competitive results in ASR over many benchmarks against multiple backdoor attacks.

+ The proposed method only needs one-layer finetuning which is efficient.

Cons:
- The motivation of NGF does not fully convince me. The authors are suggested to explain more of geometric awareness in Fisher Information Matrix and why it is helpful for smoother optimization. Is NGF applicable for general optimization where the smoother surface is supposed to be helpful for many tasks?

- The experimental results on ACC seem not good. Is that true? What is the reason for this, and how to solve it?

- The baseline methods are not very recent (ANP in 2021). Since this is for ICLR 2023, is it any baseline in 2022?


Questions/Other Comments:

1. Please consider changing the color of the references. The black references are hard to distinguish, and they may interrupt reading.

2. Could authors explain more about ASR and ACC? How to compute ASR and ACC?


**Summary Of The Paper:**

This paper presents a new method to recover the model after an attack by finetuning only one layer. Moreover, it has assumed that the backdoor model tends to be trapped in the local minimum. Therefore, a new purification technique named NGF is invented based on the loss surface curvature matrix, i.e., Fisher Information Matrix. The proposed method is evaluated on multiple benchmarks across 11 backdoor attacks with promising results in ASR.

**Summary Of The Review:**

Based on the comments above, this submission is around the borderline. I will consider changing the score after discussion.

---

> ### Author Response · Authors · 2022-11-08
> **Response to Reviewer 6Zjq (2/2)**
>
>
> **Reviewer’s Comment** *The baseline methods are not very recent (ANP in 2021). Since this is for ICLR 2023, is it any baseline in 2022?*
>
> **Our Response**
>
> We appreciate the reviewer’s concern on comparison with additional baselines. To address this concern, we have further compared our method with two very recent baselines, ICLR 2022 accepted paper *Implicit Backdoor Adversarial Unlearning (I-BAU)[1]* and the NIPS 2022 accepted paper *Adversarial weight masking (AWM)[2]*. Motivated by ANP, AWM solves a min-max optimization problem to recover the trigger pattern and then finds sensitive network weights corresponding to that pattern.  In their work, AWM also tries to address the concern of limited validation data during backdoor purification. I-BAU proposes a minimax formulation for backdoor defense and solves it utilizing the implicit hypergradient that describes the interdependence between inner (max) and outer (min) optimization. As for the training setup, we consider CIFAR10 dataset with PreActResNet-18 architecture and a poison rate of 10%. Table 2 shows that NGF outperforms both of these methods in terms of backdoor removal performance. Similar to ANP, AWM usually works well for weak attacks with low poison rates and often struggles to deliver the same level of performance under more challenging attack settings. I-BAU also underperforms as its trigger recovery algorithm may not be highly suitable in most attack scenarios. On the other hand, NGF performs consistently over diverse attack settings in a faster and more effective manner. Note that, we have considered an additional attack for this particular comparison, WaNet[3].
>
> **Table 2: Comparison of NGF with some of the recent defense techniques. We consider an additional attack WaNet[3] for this comparison. All entries in the table are in ASR/ACC format.** ***Note:*** Please refer to Table 2 of our revised paper for the full evaluation of new baseline defenses and attacks.
> |Method|Badnets  |  Blend     | Trojan   |   CLB      |   WaNet[3]  |
> |:----------|:----------|:----------|:----------|:----------|----------:|
> |No Defense  |100/92.96 | 100/94.11 | 100/89.57 | 100/92.78| 98.64/92.29|
> |AWM[1]    |4.34/86.17| 2.13/88.93| 5.41/86.45| 1.89/84.18| 6.68/86.05|
> |I-BAU[2]   |9.72/87.58| 11.53/90.84| 7.91/**87.24** | 5.78/86.70| 10.72/85.94|
> |NGF     |**1.86/88.32**| **0.38/91.17**| **2.64**/84.21 | **1.04/88.37**|  **2.38/89.58**|
>
>
> [1] "Adversarial unlearning of backdoors via implicit hypergradient." ICLR’22
>
> [2] "One-shot Neural Backdoor Erasing via Adversarial Weight Masking." NeurIPS 2022.
>
> [3] “WaNet -- Imperceptible Warping-based Backdoor Attack.” ICLR’21
>
> *****
> *****
>
> **Response to Questions/Other Comments:**
>
> 1. We change the color for references in the updated version.
>
> 2. **Formulations of ACC and ASR:**
>
> We define ACC as the test accuracy on the clean test dataset. More plainly, ACC is the ratio of clean test samples predicted as their ground truths.
> On the other hand, attack success rate (ASR) can be defined as the misclassification rate of triggered test samples to the adversary-set target label. ASR is the ratio of samples predicted as the target label rather than their original ground truth. We add attack-specific triggers to each test sample and compare their predicted label to the target label. ASR should be close to 0% for a successful defense method.
>
> ***
> ***
> We thank the reviewer again. We aimed to address the reviewer’s valuable comments and provided additional clarification behind the intuition of our proposed method. We would be happy to address any further concerns.

---

> > ### Author Response · Authors · 2022-12-01
> > **Follow Up. Thank You for Your Review Again!**
> >
> > Dear Reviewer,
> >
> >
> >  We genuinely appreciate your positive feedback once again! Your constructive feedback properly guided us to revise the paper and made it much better. This is a friendly reminder that the final stage of the discussion period is drawing close. We would really appreciate it if you could let us know whether your concerns have been addressed and if we can clarify anything else.
> >
> >
> > Nevertheless, Thank you for your kind efforts :),
> >
> > Authors of Paper 2286

---

> ### Author Response · Authors · 2022-11-08
> **Response to Reviewer 6Zjq (1/2)**
>
> Thank you for the positive comments and valuable feedback. We addressed the reviewers concerns as follows:
>
> **Reviewer’s Comment:** *The motivation of NGF does not fully convince me. The authors are suggested to explain more of geometric awareness in Fisher Information Matrix and why it is helpful for smoother optimization. Is NGF applicable for general optimization where the smoother surface is supposed to be helpful for many tasks?*
>
> **Our Response**
> The reviewer pointed out an important aspect on the ability of FIM-based optimizer in model smoothness. While we have addressed this in the original submission, we try to elaborate more here.
>
> *More explanation on Fisher Information Matrix-based optimization* In Fisher Information Matrix (FIM) based optimization, the natural gradient is defined as $F^{-1} \nabla L$ (ref. Eq. (5) of our main paper) [1]. From the perspective of information geometry [2], natural gradient defines the *direction in parameter space*  which gives largest change in objective *per unit of change **in model ($p(y|x,\theta)$)***. Per unit of change in model is measured by KL-divergence. Note that KL-divergence is well connected with FIM as it can be used as a local quadrature approximation of  KL-divergence of *model change*. The equation FIM (Eq. (2) in our paper) suggests that one requires the knowledge of the original parameter ($\theta$) space to estimate it. Therefore, FIM can be thought of as a mechanism to translate between the geometry of the model ($p(y|x,\theta)$) and the current parameters ($\theta$) of the model. The way natural gradient defined the *direction in parameter space* is contrastive to the stochastic gradient. Stochastic gradient defines the direction in parameter space for largest change in objective *per unit of change **in parameter ($\theta$)*** measured by Eucludian distance. That is, the gradient direction is solely calculated based on the changes of parameters, without any knowledge of model geometry.
>
> As FIM-based optimization minimizes the *changes in model*, the model itself cannot significantly change at each iteration. So, the overall optimization process goes through comparatively smoother transition and finally reaches smoother minima in comparison with SGD-based optimization.
>
>
> *Is NGF applicable to general optimization?* NGF merely fine-tunes the last layer (classifier layer) of a pre-trained model–it uses an unmodified CNN backbone (feature extractor). Therefore, the current version of NGF cannot be directly applied to optimize a model from scratch. However, NGF can be adopted for ‘transfer learning’ tasks where the regularization term can be dropped depending on the task. In some scenarios, using a pretrained model on a new task may require changing only couple of layers. In that scenario, NGF can be used to achieve smoother minima and an improved generalization performance.
>
>
> [1] Shun-Ichi Amari. “Natural gradient works efficiently in learning.” Neural computation, 1998.
>
> [2] James Martens. “New insights and perspectives on the natural gradient method.” The Journal of Machine Learning Research, 2020.
>
>
>
> ****
>
> **Reviewer’s Comment:** *The experimental results on ACC seem not good. Is that true? What is the reason for this, and how to solve it?*
>
> **Our Response:**
>
> We thank the reviewer for raising this important concern. We assume the reviewer mentions the clean test accuracy (ACC) of a backdoor model. Usually, a benign and a backdoor model do not have the same level of ACC; but ACC should be close. Creating a backdoor model with the same accuracy (ACC) as the benign model is very challenging. This is due to the fact that the same number of clean samples are not available for both scenarios. Furthermore, the model has to learn features from both the clean and triggered samples. Therefore, compared to a benign model, a backdoor model has to learn two tasks. Even though a deep neural network is very powerful at learning useful representations, an additional task (backdoor insertion) for the backdoor model limits its generalization performance (ACC) on the clean samples. For some attacks, e.g. TrojanNet, the ACC becomes lower than usual while performing a strong and successful attack.
>
> As for the solution to this problem, we can decrease the poison rate. To this end, we aim to create a backdoor model with a low ACC drop by setting the poison rate to 1%. The closest backdoor model we get is for the Blend attack on CIFAR10 dataset (95.04% vs 95.21%). We report the performance of NGF for this particular scenario too in Table 1.
>
> **Table 1: Performance of NGF with 1\% poison rate. Compared to the benign model, all backdoor models still have ACC drop although Blend model is very close to the benign model. We use CIFAR10 dataet here and all entries in the table are in ASR/ACC format.**
> |Defense Method|Benign|Badnets|Blend |Trojan |
> |:-|:-|:-|:-|-:|
> |No Defense|0/95.21|100/93.82|100/95.04|100/91.11|
> |NGF|0/94.10|1.08/90.23|0.51/93.61|1.96/88.12|

---

### Official Review · Reviewer_uefD · 2022-10-26

**Confidence:** 4
**Correctness:** 3
**Technical Novelty And Significance:** 2
**Empirical Novelty And Significance:** 3
**Recommendation:** 6

**Clarity, Quality, Novelty And Reproducibility:**

The implementation is well supported by experiments.

There are some typos in the text, such as "Eigenvlaue" should be "Eigenvalue" in Figure 1, the last ")" in eqn(3) is missing, and so on.

**Strength And Weaknesses:**

Strength:
1. The motivation of this paper is clear, and the experiments are effective in proving the proposition of removing Backdoor from the model by smoothing the loss surface of the model.
2. This paper demonstrates the effectiveness of the proposed approach through extensive and comprehensive experiments. The experimental results under various Backdoor Attack methods and models are included, and the effectiveness and superiority of the proposed approach is demonstrated by comparing various mainstream defense methods.

Weaknesses:
1. Not much intuition and discussion on how the proposed method will help eliminate the non-smoothness and why it is better than other solutions.
2. lack comparison with new defense baselines
3. technical contribution is not much aside from a regularization term in loss and the use of NGD


**Summary Of The Paper:**

This paper proposed NGF, which uses clean data to remove the backdoor from a model.  The proposed method works by fine-tuning the last layer of the model such that the false local minima are smoothed out and the effect of removing the backdoor from the model is achieved. Finally, this paper gives very extensive and comprehensive experiments to demonstrate the performance of their proposed model


**Summary Of The Review:**

1. There lacks sufficient intuition on why smoothness is the key to remove the backdoor.  Essentially smoothness measures the changes in gradients wrt to changes in inputs. Then the authors’ conjecture is that triggers in input could lead to large changes in gradients. But why it must be a backdoor? It seems to me that the lipshitz pruning method (CLP, Zheng et al. 2022) is telling a more convincing story that if triggers in input could lead to large changes in loss, it might be backdoored.


2. For runtime comparison, the author should compare with other baselines as well as NGF without regularization term. Also for ablation study, I would like to see the results for fine-tuning on all layers.

3. The proposed method requires to have clean training data in order to function properly. The authors did not test the situation when the number of clean training data is not sufficient. To be comprehensive, I would suggest the authors to include the number of training samples as an ablation study.

4. The performance reported for data-free backdoor removal by Zheng et al. 2022 seems a bit strange as the original paper actually gives quite a good performance on Blend, TrojanNet. Did the author tune the hyperparameters?

5. There is also some recent work on backdoor removal:

    "One-shot Neural Backdoor Erasing via Adversarial Weight Masking." NeurIPS 2022.

    The authors may want to comment on/compare with the above work.

---

> ### Author Response · Authors · 2022-11-08
> **Response to Reviewer uefD (2/2)**
>
> **Reviewer’s Comment:** *There is also some recent work on backdoor removal: "One-shot Neural Backdoor Erasing via Adversarial Weight Masking." NeurIPS 2022. The authors may want to comment on/compare with the above work*
>
>  **Our Response:**
> We appreciate the reviewer's guide for an additional baseline. To address this comment, we have further compared our method with two very recent baselines: the reviewer’s suggested paper *Adversarial weight masking (AWM)[1]* (NeurIPS 2022) and ICLR 2022 accepted paper *Implicit Backdoor Adversarial Unlearning (I-BAU)[2]*. Motivated by ANP, AWM solves a min-max optimization problem to recover the trigger pattern and then finds sensitive network weights corresponding to that pattern.  In their work, AWM also tries to address the concern of limited validation data during backdoor purification. I-BAU proposes a minimax formulation for backdoor defense and solves it utilizing the implicit hypergradient that describes the interdependence between inner (max) and outer (min) optimization. As for the training setup, we consider CIFAR10 dataset with PreActResNet-18 architecture and a poison rate of 10%. Table 1 shows that NGF outperforms both of these methods in terms of backdoor removal performance. Similar to ANP, AWM usually works well for weak attacks with low poison rates and often struggles to deliver the same level of performance under more challenging attack settings. I-BAU also underperforms as its trigger recovery algorithm may not be highly suitable in most attack scenarios. On the other hand, NGF performs consistently over diverse attack settings in a faster and more effective manner. Note that, we have considered an additional attack for this particular comparison, WaNet[3].
>
> **Table 1: Comparison of NGF with some of the recent defense techniques. We consider an additional attack WaNet[3] for this comparison. All entries in the table are in ASR/ACC format.** ***Note:*** Please refer to Table 2 of our revised paper for the full evaluation of new baseline defenses and attacks.
> |Method|Badnets  |  Blend     | Troj-one   |   CLB      |   WaNet[3]  |
> |:----------|:----------|:----------|:----------|:----------|----------:|
> |No Defense  |100/92.96 | 100/94.11 | 100/89.57 | 100/92.78| 98.64/92.29|
> |AWM[1]    |4.34/86.17| 2.13/88.93| 5.41/86.45| 1.89/84.18| 6.68/86.05|
> |I-BAU[2]   |9.72/87.58| 11.53/90.84| 7.91/**87.24** | 5.78/86.70| 10.72/85.94|
> |NGF     |**1.86/88.32**| **0.38/91.17**| **2.64**/84.21 | **1.04/88.37**|  **2.38/89.58**|
>
>
> [1] "One-shot Neural Backdoor Erasing via Adversarial Weight Masking." NeurIPS 2022.
>
> [2] "Adversarial unlearning of backdoors via implicit hypergradient." ICLR’22
>
> [3] “WaNet -- Imperceptible Warping-based Backdoor Attack.” ICLR’21
>
> ***
>
> **Reviewer’s Comment:** *The authors did not test the situation when the number of clean training data is not sufficient. To be comprehensive, I would suggest the authors to include the number of training samples as an ablation study’*
>
> **Our Response:**
> Thanks for the comment. As per the reviewers' suggestion, we report results with different validation data sizes. We kindly request the reviewer to check Appendix J (Table 14). Our method performs well even with 50 validation data points. We have compared with ANP and AWM[1].
>
> [1] "One-shot Neural Backdoor Erasing via Adversarial Weight Masking." NeurIPS 2022.
>
> ***
> ***
>
> We thank the reviewer again for the valuable feedback. We hope our responses make the reviewer’s concerns clear. We would be happy to provide further explanation if the reviewer still has some concerns.

---

> ### Author Response · Authors · 2022-11-08
> **Response to Reviewer uefD (1/2)**
>
> We thank the reviewer for valuable feedback. The reviewer pointed out several important aspects and concerns about our work which we addressed as follows:
>
> **Reviewer’s Comment:** *There lacks sufficient intuition on why smoothness is the key to removing the backdoor. Essentially smoothness measures the changes in gradients wrt to changes in inputs. Then the authors’ conjecture is that triggers in input could lead to large changes in gradients. But why it must be a backdoor? It seems to me that the Lipschitz pruning method (CLP, Zheng et al. 2022) is telling a more w2convincing story that if triggers in input could lead to large changes in loss, it might be backdoored.*
>
> **Our Response**
>
> *Why smoothness is the key to removing the backdoor:* One key observation from the smoothness study is that: there exists a key difference between weight-loss surface smoothness (estimated by *loss hessian*) of a backdoor and a benign model w.r.t. clean distribution—the weight-loss surface of a backdoor model is less smooth compared to a benign model. To further elaborate, let us consider feeding a clean sample to a backdoor model. By definition, it will predict the correct ground truth label. Now, consider feeding a sample with a backdoor trigger on it. The model will predict the adversary-set target label implying significant changes in prediction distribution. This significant change can be explained by surface smoothness. In order to accommodate this significant change in prediction, the model must adjust itself accordingly. Such adjustment leads to non-smoothness in the weight-loss surface. The reviewer's intuition of gradient change aligns with our intuition. A non-smooth surface causes significant changes in *loss gradient*  for specific inputs.  In our case, these specific inputs are backdoor-triggered samples.  As the magnitude of a trigger is usually very small compared to the total input magnitude, the model has to experience quite a significant change in its weight space to cause large loss changes. We characterize this change in terms of smoothness. As for backdoor removal, we claim that making the non-smooth weight loss surface smoother removes the backdoor behavior. Based on the above discussion, a smoother surface should not cause a large change in loss or model predictions corresponding to backdoor-related perturbations or triggers.  In summary, for a model to show certain backdoor behavior, there are some specific changes that take place in the weight space. In this work, we try to explain these changes in terms of weight-loss surface smoothness. Our intuition is well supported by our comprehensive empirical evaluations.
>
> *Intuition behind CLP (Zheng et al. 2022) vs NGF:* The reasonings for the presence of potential backdoor triggers in input sample are essentially similar in both CLP and our method (NGF)—large changes in loss (equivalently, large changes in gradient to the corresponding loss). However, the metrics to capture such behavior are different: Lipshitz constraint in CLP and Smoothness in NGF. While minimizing Lipshitz constraint (in CLP) bounds the changes of gradient, smoothness (in NGF) helps to tolerate the presence of a backdoor trigger in input without *significantly* changing the loss (equivalently, the prediction distribution).
> ****
>
> **Reviewer’s Comment:**  *For runtime comparison, the author should compare with other baselines as well as NGF without regularization terms. Also for the ablation study, I would like to see the results for fine-tuning on all layers.*
>
> **Our Response:**
> For runtime analysis, we have already compared our method with other baselines in our original submission (in Table 11 of Appendix H). Our method offers better runtime performance. We have updated Table 11 with the runtime analysis of NGF without the regularizer.
> Vanilla FT indicates fine-tuning with all layers with SGD optimizer. We have reported results for Vanilla FT in Table 2. Per the reviewer’s suggestions, we have added further evaluations in Appendix J (Table 12) when we fine-tune all layers with NGF and Vanilla FT (SGD).
>
>
> ***
> **Reviewer’s Comment:** *The performance reported for data-free backdoor removal by Zheng et al. 2022 seems a bit strange as the original paper actually gives quite a good performance on Blend, TrojanNet. Did the author tune the hyperparameters?*
>
> **Our Response:**
> We used the original source code provided by Zheng et al. 2022 as it is; without any hyper-parameter tuning on our part. Moreover, even without considering the results of Blend and TrojanNet, Table 9  in Appendix E shows that NGF outperforms Zheng et al. 2022 for other attack benchmarks. Pruning backdoor-affected channels usually provide good performance for backdoor removal (low ASR) but may suffer from low clean test accuracy as many neurons (that may be responsible for ACC) are pruned at the same time.

---

### Public Comment · ~Zhiwei_Jia1 · 2022-11-11
**Related Work**

Hi authors, I really like your work on using the Fisher Information Matrix to improve the robustness of DNN. Please consider citing the following paper that also utilizes FIM for better generalization.

[1] Information-Theoretic Local Minima Characterization and Regularization

ICML 2020

Zhiwei Jia, Hao Su

---

### Author Response · Authors · 2022-11-14
**Updates on the revision**

We thank all the reviewers for their valuable comments and insightful feedback on our paper. We carefully read the reviewers' comments and believed that addressing those comments would improve the paper significantly. We have revised the main paper as well as the appendices and tried to include all the reviewers' comments and suggestions in the revision. Specifically, we have

I. added new baseline defenses and attacks in the evaluation section (in Table 2).

II. added a new Appendix (Section C) with more explanations behind our motivations, and tried to address the following concerns: 'why smoothness is key to removing backdoor?', 'How FIM helps to achieve smoothness', and 'Why SAM does not work in the proposed problem setup'.

III. added an ablation study on fine-tuning all layers with NGF (in Appendix J).

We have also revised several other sections following reviewers' comments and added pointers to those updates in the reviewer-specific responses. We thank the reviewers again and hope that our response could address all questions and concerns. We would be happy to revise our paper again if the reviewers have more concerns.

**Note:** We have highlighted the changes in revision using the 'Cyan' color.

---

### Author Response · Authors · 2022-11-25
**General Response**

Dear ACs and Reviewers,


Thank you for your valuable efforts in the reviewing process.

We sincerely hope that we could address most of the concerns raised by the reviewers and would be happy to have further discussions on any remaining questions the reviewers may have about the paper. As the deadline approaches, let us know if you require further clarification on any of our responses.

We appreciate the time and energy put in by the reviewers. If you could kindly go through our responses and the changes in the paper, we would really appreciate it.


Sincerely,

The Authors

---

### Decision · Program_Chairs · 2023-01-20

**Decision:**

Reject

**Justification For Why Not Higher Score:**

see above

**Justification For Why Not Lower Score:**

N/A

**Metareview: Summary, Strengths And Weaknesses:**

This work proposed to mitigate the backdoor effect by fine-tuning the last layer (linear classifier) using Natural Gradient Descent (NGD), to find a smooth minimum, corresponding to a good model (without backdoor).

The initial scores of this work are 555, and there are several important concerns, including: 1) the motivation of the connection between smoothness and backdoor; 2) the limited novelty, as just NGD together with FIM, without theoretical analysis; 3) difference with CLP; 4) inadequate experiments.

The authors made significant efforts in the rebuttal by providing some explanations and additional results. Then the scores are raised to 666. The main reasons of no higher scores are:
1) The motivation of the connection between smoothness and backdoor is not well theoretically verified, though it seems to be reasonable. The novelty of the proposed method NGD with FIM is technically limited, and without theoretical analysis. (One more word, the guessed reason of why SAM doesn't work are not verified. Actually it could be empirically verified by explicitly measuring the smoothness of the model fine-tuned by SAM and the proposed method.)
2) The general idea is similar to CLP, both methods aimed to control the change of loss, one using smoothness, while one using Lipschitz. Moreover, CLP is data-free defense, while this work requires clean data.

I thoroughly read the paper, reviews, rebuttals, and the revised manuscript.  And, I carefully check the reported experimental results, and have the following two comments:
1) The main results are shown in Table 2. But it is very strange that on different datasets, the evaluated attack methods are very different. It reveals that the authors carefully chose the results that advocate the proposed method, while some bad results of the proposed method are hidden. I didn't see any reasonable reason about this setting (for example, the original papers of some compared methods reported evaluations on some datasets, but missing here), and didn't find any explanation from the manuscript.

2) The comparison with CLP is most important, considering the high similarity. The authors added incomplete comparisons with CLP in appendix, only on CIFAR-10. The authors claimed that although CLP could remove the backdoor very well, but the clean accuracy on some attacks drops significantly. I carefully compare the reported results in this manuscript and the original paper of CLP, and find significant conflicts. Let's compare Table 9 in this work and Table 3 of CLP (https://www.ecva.net/papers/eccv_2022/papers_ECCV/papers/136650171.pdf), with same dataset CIFAR-10 and same network ResNet-18. For the same attack methods BadNets, Blended, Trojan, the reported results of CLP from this work are much much worse than the results from original CLP paper. Obviously, the reported results of CLP in this work are WRONG.
Moreover, remember that CLP is data-free, while this work requires clean data. Let's compare Table 9 and Table 14 in the revised manuscript, it can be found that only when the clean data achieves 500, its performance is slightly better than the reported CLP in Table 9. I  guess, if the CLP model is further fine-tuned based on the same clean data, its performance (especially the clean accuracy) could be further improved in Table 9 (even I assume the reported results are reliable).

Thus, the reported results are highly unconvincing.

In summary, although this work provides some insight about backdoor, there is lack of theoretical analysis. What is more important, the experimental results are unconvincing.

---

> ### Author Response · Authors · 2023-02-01
> **Concerns to Meta-Reviewing**
>
> Dear Program Chairs,
>
> We respectfully disagree with the meta-reviewer about these observations. Clearly, the meta-reviewer (MR) needs to be better informed on this topic.
>
>
> **“The main results are shown in Table 2. But it is very strange that on different datasets, the evaluated attack methods are very different. It reveals that the authors carefully chose the results that advocate the proposed method, while some bad results of the proposed method are hidden. I didn't see any reasonable reason about this setting (for example, the original papers of some compared methods reported evaluations on some datasets, but missing here), and didn't find any explanation from the manuscript”**--This is an easily addressable comment which (we don’t believe) could be a reason to reject a paper! It is very difficult to implement attacks like Dynamic, BPPA for tiny-ImageNet, and ImageNet and get good attack success rates. An attack without high attack success rates is not considered a successful attack. And if an attack is not successful, what is the point of defending it? We did not report results for these attacks because we could not create successful attacks (using Dynamic and BPPA techniques) for these two datasets. Besides, very few defenses up to now considered datasets other than CIFAR10 and GTSRB. So what is the point of hiding results for ImageNet? We consistently got an improvement for those two datasets that AWM, I-BAU, and ANP reported their results on. How is it that a downside of our method? However, we could have easily addressed this concern in the revised version.
>
>
> **”Obviously, the reported results of CLP in this work are WRONG:”**  We strongly disagree with the MR’s judgment. In following, we addressed MR’s specific concern:
>
> **"methods should have the same results for same dataset CIFAR-10, and same network ResNet-18"**--This is an absolutely wrong statement. Final Results do not simply depend on the dataset and network model, rather it depends on how one trained an attack with the dataset on the network model. Note that, “attack strengths such as trigger properties (size, color, shape, etc.), trigger ratio, clean training data size, etc.” are very critical for attack implementation. It is highly possible CLP has verified its method under limited settings where they generated weaker backdoors. Please note that CLP did not release their source code for their attack implementation.
>
>
> *Considering the meta-reviewer's argument to be logical (although they are not!), why did not the MR bring them forward during the discussion period? What is the real purpose of the discussion phase then?* You make the authors and reviewers go back and forth and waste their valuable time and energy only to make a **last-minute decision**. We strongly believe the program chairs will reconsider the meta-reviews considering that most of these concerns, specifically results related to CLP discussed in the rebuttal following the Reviewer uefD’s comment. Finally, no reviewer has any major concerns about the paper.
>
> Thanks,
>
> Paper 2286 Authors